# Atmospheric bias teleconnections in boreal winter associated with systematic sea surface temperature errors in the tropical Indian Ocean

**Yuan-Bing Zhao, Nedjeljka Žagar, Frank Lunkeit, and Richard Blender**

Meteorologisches Institut, Universität Hamburg, Hamburg, Germany

**Correspondence:** Yuan-Bing Zhao (yuan-bing.zhao@uni-hamburg.de)

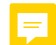

**Abstract.** Coupled climate models suffer from significant sea surface temperature (SST) biases in the tropical Indian Ocean (TIO), leading to errors in global climate predictions. In this study, we investigate atmospheric bias teleconnections caused by the TIO SST bias and their impacts on model variability. A set of century-long simulations forced by idealized SST perturbations, which resemble various (monopolar or dipolar, positive or negative) persistent TIO SST biases in coupled climate models, are conducted with an intermediate-complexity atmospheric model. Bias teleconnections with a focus on boreal wintertime are researched using the normal-mode function decomposition, which can differentiate between balanced and unbalanced flow regimes across spatial scales. The results show that the atmospheric circulation biases caused by the TIO SST bias have the Gill–Matsuno-type pattern in the tropics and Rossby-wave-train distribution in the extratropics, like the steady-state response to tropical heating perturbations. The teleconnections between tropical and extratropical biases are set up by Rossby wave activity flux emanating from the subtropics. Over 90 % of the bias variance is contained in zonal wavenumbers $k \leq 5$. Comparisons among experiments show that the northward shift of the SST bias away from the Equator weakens the atmospheric response, but it does not change its structure. Besides, the positive SST bias produces stronger bias teleconnections than the negative one of the same size and magnitude.

The response of the spatial variance (i.e. energy) to the TIO SST biases depends on dynamical regimes and spatial scales. Across all experiments, the unbalanced zonal-mean flow energy decreases, whereas its balanced counterpart increases. These changes primarily arise from the strong covariance between the circulation bias and the reference state (i.e. bias covariance). For the wave flow energy, the unbalanced and the tropical balanced parts show a synchronized response. Both increase in experiments with monopolar SST bias and decrease in that with dipolar SST bias. The increase is mainly due to the bias variance, whereas the decrease is due to a strong negative bias covariance at zonal wavenumbers $k = 1$ and 2. In contrast, the extratropical balanced wave flow energy decreases in experiments with positive SST bias (including the dipolar case) and increases in that with the negative SST bias, which is mainly due to the bias covariance at zonal wavenumber $k = 1$. The response of the interannual variance (IAV) is contingent upon the sign of the SST bias. A positive SST bias reduces the IAV, whereas a negative one increases it, regardless of dynamical regimes. Geographically, strong IAV responses are observed in the tropical Indo-West Pacific region, Australia, South and Northeast Asia, the Pacific–North America region, and Europe, where the background IAVs are strong.

## 1 Introduction

The tropical Indian Ocean (TIO) plays an important role in climate variability (e.g. Schott et al., 2009; Beal et al., 2020; Hermes et al., 2019). A realistic representation of the TIO sea surface temperature (SST) in the ocean–atmosphere coupled general circulation models (CGCMs) is crucial for the accurate prediction of both local and remote climate. However, state-of-the-art CGCMs exhibit large systematic errors (biases) in the TIO SST (Li and Xie, 2012), which inevitably

affect the accuracy of the atmospheric simulations (Joseph et al., 2012; Levine and Turner, 2012; Prodhomme et al., 2014). This study aims at understanding the local and, more importantly, the remote effects of the TIO SST biases, which are
5 termed "bias teleconnections".

The TIO SST biases vary widely in amplitude and sign among the models involved in the Coupled Model Intercomparison Project (CMIP). About half of the CMIP5 models show positive SST biases over the western TIO during bo-
10 real summer (Fathrio et al., 2017; Lyon, 2020). For instance, the SINTEX-F2 model has a warm SST bias in the tropics, especially over the western TIO region (Joseph et al., 2012), and this bias is nearly constant throughout the annual cycle (Prodhomme et al., 2014). Other half of the CMIP5
models show negative SST biases over the entire TIO region throughout the year (Wang et al., 2014; Fathrio et al., 2017). For instance, the HadCM3 model has a large-magnitude, cold SST bias in winter and spring in the Arabian Sea (Turner et al., 2005; Levine and Turner, 2012). Similar cold bias is
present in most CMIP6 models (Wang et al., 2022). In addition, a significant warm TIO SST bias was recently reported in the CMIP6 models, which was not evident in the CMIP5 models (Zhang et al., 2023). Furthermore, the SST biases are also coupled with the representation of the Indian
Ocean Dipole (IOD), which remains overly strong in all generations of CMIP models (Cai and Cowan, 2013; Weller and Cai, 2013). The IOD-like SST biases in the majority of the CMIP5 models (Li et al., 2015) are shown to be even stronger in the CMIP6 models (McKenna et al., 2020).

The long-standing TIO SST biases impact the skill of climate prediction. Locally, the TIO SST biases modify the meridional SST gradient and affect the Indian summer monsoon, leading to biases in precipitation (Joseph et al., 2012; Levine and Turner, 2012; Bollasina and Ming, 2013; An-
namalai et al., 2017; Prodhomme et al., 2014). Moreover, an SST increase from 26.5 to 28.0 °C has been shown by both observations and simulations to drastically change the convective response from shallow to deep convection (e.g. Gadgil et al., 1984; Roxy et al., 2014). On the other hand,
SST biases also have remote impacts. Wang et al. (2014) found in CMIP5 that local SST biases can be linked with those in far away regions. Recently, Stan et al. (2023) and Bai et al. (2023) showed that the tropical West Pacific warm bias in the Unified Forecast System global coupled
model can influence the simulated North American precipitation remotely through the wave activity flux.

However, the impacts of TIO SST biases on simulated atmospheric variability are more difficult to quantify as it requires isolating the effects of the TIO SST biases from
50 other biases and the internal variability. One way to address this challenge is to perform idealized numerical experiments using the perfect-atmosphere-model framework (e.g. Žagar et al., 2020). Such experiments by Annamalai et al. (2007) showed that the diabatic heating associated with the TIO sea-
55 sonally varying SST anomalies can generate Rossby wave packets which set up the teleconnection between the TIO and the Pacific–North American (PNA) region and considerably impact the Northern Hemisphere (NH) extratropical circulation.

The present study extends existing research of the effects
of systematic errors in the TIO SST on the atmospheric variability by quantifying spatial and temporal variance changes from a global perspective. We hypothesize that TIO SST biases induce biases in the simulated atmospheric circulation not only in the tropics but also in the extratropics through
tropics–extratropics coupling and that TIO SST biases affect the spatio-temporal variability across scales. Specific questions to be addressed in this paper are as follows:

– What is the spatial structure of biases in simulated circulation in the tropics and extratropics due to the TIO
SST biases? Which zonal scales are most affected?

– What are the dynamical mechanisms that connect the TIO SST bias to biases in simulated circulation?

– How do global spatial and temporal variability change in response to the TIO SST biases?

We adopt the perfect-atmosphere-model framework with an intermediate complexity climate model (Planet Simulator; Fraedrich et al., 2005), which allows us to easily interpret the model results. A reference (or control) simulation forced by observed SST is conducted and compared to sensitivity sim-
80 ulations forced by the same SST but augmented with SST biases in the TIO region. The scale and regime decomposition of the circulation is performed using the normal-mode function decomposition (Žagar et al., 2015). The method for studying the effects of biases on simulated variability in cli-
85 mate models was developed by Žagar et al. (2020). They showed that SST errors lead to large biases in circulation and interannual variability even though the atmospheric part of their CGCM is perfect. However, they studied only the total atmospheric response to globally distributed SST errors,
without regime decomposition. Application of the method to CMIP6 models by Castanheira and Marques (2022) showed that the presence of the cold SST bias in North Pacific may compensate for the biases in the North Pacific barotropic atmospheric variability through influence on the excitation of
the most unstable barotropic mode of the atmospheric circulation.

The rest of this paper is organized as follows. In Sect. 2, we describe the Planet Simulator and the experiment design, followed by the description of methods for the quantification
of circulation and variability biases. The results are presented in Sect. 3, including the total circulation biases, the regime-dependent biases, the underlying physics of remote bias coupling, and its impacts on the simulated spatio-temporal variability. The study is summarized in Sect. 4.

## 2 Methodology

We first present the design of numerical experiments with the Planet Simulator (PLASIM; Fraedrich et al., 2005; Fraedrich and Lunkeit, 2008). This is followed by the description of the normal-mode function (NMF) decomposition and definition of quantities analysed in modal and physical space.

### 2.1 Climate model and experiment design

#### 2.1.1 The model

PLASIM models the atmospheric dynamics using primitive equations in the $\sigma$ coordinate, $\sigma = p/p_s$, where $p$ and $p_s$ denote the pressure and the surface pressure, respectively. The prognostic variables are vorticity, divergence, temperature, specific humidity, and the logarithm of the surface pressure. The hydrostatic primitive equations are solved using the spectral transformation methods (Eliassen et al., 1970; Orszag, 1970). Unresolved processes are parameterized, which include the long-wave (Sasamori, 1968) and short-wave (Lacis and Hansen, 1974) radiation, interactive clouds (Stephens, 1978, 1984; Slingo and Slingo, 1991), moist (Kuo, 1965, 1974) and dry convection, large-scale precipitation, the horizontal and vertical diffusion (Louis, 1979; Louis et al., 1982; Laursen and Eliasen, 1989; Roeckner et al., 1992), and boundary-layer parameterization including latent and sensible heat fluxes (Louis, 1979). For more information on the model, the reader is referred to Fraedrich et al. (2005).

In this study, we run PLASIM with prescribed SST and sea ice content. We adopt a T31 horizontal resolution (approximately $3.75° \times 3.75°$ on the corresponding $96 \times 48$ Gaussian grid) and $10\,\sigma$ levels: 0.038, 0.119, 0.211, 0.317, 0.437, 0.567, 0.699, 0.823, 0.924, and 0.983.

The atmosphere–ocean coupling plays a crucial role in the simulation. With the SST fixed, the atmosphere and ocean surface communicate by exchanging the heat and moisture. The bulk aerodynamic formulas for the surface fluxes are

$$F_T = \rho c_p C_H |V|(\text{SST} - T_{\text{air}}),\qquad(1)$$

$$F_q = \rho C_H C_W |V|(q_{\text{sea}} - q_{\text{air}}),\qquad(2)$$

where $F_T$ and $F_q$ are the sensible heat flux and the moisture flux, respectively. The constant parameters in the formulas are defined as follows: $\rho$ is air density, $c_p$ is the specific heat at constant pressure, $C_H$ is the transfer coefficient for heat, and $C_W$ is the wetness factor accounting for different evaporation efficiencies due to surface characteristics. The fluxes depend on the horizontal wind speed ($|V|$), air temperature ($T_{\text{air}}$) and specific humidity ($q_{\text{air}}$) at the lowermost model level, and the specific humidity at sea surface ($q_{\text{sea}}$), which is assumed to be saturated and has a temperature equal to the SST. Given the SST, the saturation water vapour pressure ($e_{\text{sat}}$) is computed using the Clausius–Clapeyron equation for SST (in °C) as

$$e_{\text{sat}} = 610.78 \exp\left(\frac{17.27\text{SST}}{\text{SST} + 237.30}\right).\qquad(3)$$

Then $q_{\text{sea}}$ is produced, making use of instantaneous value of surface pressure $p_s$ as

$$q_{\text{sea}} = \frac{\epsilon e_{\text{sat}}}{p_s - (1 - \epsilon)e_{\text{sat}}},\qquad(4)$$

where $\epsilon$ is the ratio of the gas constant for dry air ($R_d$) and water vapour ($R_v$) ($\epsilon = R_d/R_v$). A positive $F_T$ means heat is flowing from the surface to the atmosphere. A positive $F_q$ means water is evaporating from the surface with latent heat. Both terms exhibit seasonal variations. The globally averaged climatological annual mean $F_T$ in PLASIM forced with SST from the ERA-20C reanalyses (Poli et al., 2016) is estimated to be $20.5\,\text{W}\,\text{m}^{-2}$, and the seasonal variation of $F_T$ spans from $-14.2\,\%$ to $15.4\,\%$ of the annual mean. The globally averaged climatological annual mean $F_q$ is measured at $2.8\,\text{kg}\,\text{m}^{-2}\,\text{d}^{-1}$, corresponding to a latent heat flux of $81.2\,\text{W}\,\text{m}^{-2}$, and the seasonal variation of $F_q$ ranges from $-4.4\,\%$ to $6.6\,\%$ of the annual mean.

The fluxes at the sea surface are closely coupled with the cumulus convection. In PLASIM, convection is parameterized by a Kuo-type scheme (Kuo, 1974), where an important quantity involved is the net precipitation rate, which is proportional to the net amount of the horizontal moisture flux convergence (MFC) plus the moisture supply by surface evaporation (i.e. $F_q$):

$$P_r = -\frac{p_s}{g}\int_0^1 \nabla \cdot (qV)\mathrm{d}\sigma + F_q.\qquad(5)$$

The above equations show that a systematic local increase in SST will lead to locally more upward sensible heat and moisture flux, which will lower the near-surface moist static stability. Stronger convection and more precipitation ($P_r$) will occur, leading to greater latent heat release and stronger effects on atmospheric circulation.

#### 2.1.2 Experiment design

A set of century-long simulations are carried out using a perfect-model framework. In this framework, the only difference between the reference and sensitivity simulations is the SST perturbation in the Indian Ocean, which represents the climate model SST bias. Differences between the simulations therefore originate from differences in the TIO SST.

The reference simulation is forced with the time-varying monthly mean SST, including interannual variations from the ERA-20C reanalyses (Poli et al., 2016). Four sensitivity experiments apply the same SST with the addition of time-constant perturbations in the TIO region that mimic TIO SST biases found in the CGCMs (e.g. Li et al., 2015; Fathrio et al.,

**Table 1.** Configurations of the SST biases.

| Experiments | $T_0$ (in K) | $\lambda_0$ | $\varphi_0$ | $\sigma_x$ | $\sigma_y$ |
|---|---|---|---|---|---|
| EXP_POS | 1.5 | 70° E | 0° | 20 | 15 |
| EXP_NEG | −1.5 | 70° E | 0° | 20 | 15 |
| EXP_10N | 1.5 | 70° E | 10° N | 20 | 15 |
| EXP_IOD | 1.5/−1.5 | 55° E/105° E | 0° | 15 | 15 |

2017; Lyon, 2020). In other words, the steady SST perturbations in this study represent the SST biases (or systematic errors).

The spatial structure of the SST perturbation is given by CE1

$$\Delta \mathrm{SST}(\lambda, \varphi) = \begin{cases} T_0 \exp\left[-\frac{(\lambda - \lambda_0)^2}{\sigma_x{}^2} - \frac{(\varphi - \varphi_0)^2}{\sigma_y{}^2}\right]; & \text{otherwise } |\Delta \mathrm{SST}| > 0.05, \\ 0, \end{cases} \quad (6)$$

where $T_0$ determines the maximal magnitude, $(\lambda_0, \phi_0)$ specifies the longitude and latitude of the centre location respectively, and $\sigma_x$ and $\sigma_y$ define the perturbation extent. The parameters of the four perturbations are listed in Table 1.

The distributions of SST perturbations are shown in Fig. 1. In one experiment, a monopolar SST perturbation with a maximum of $+1.5$ K is centred at the Equator and covers the entire TIO region (Fig. 1a). Another one applies the same SST perturbation but with the opposite sign (i.e. monopolar negative SST bias; Fig. 1b). For convenience, we refer to this experiment as EXP_NEG and the previous one as EXP_POS. The third experiment is the same as EXP_POS, but the SST perturbation is shifted 10° northward, and the experiment is labelled as EXP_10N (Fig. 1c). The fourth simulation applies a dipolar SST perturbation which mimics the IOD-type bias (e.g. Cai and Cowan, 2013; McKenna et al., 2020) and is referred to as EXP_IOD (Fig. 1d). In the equatorial Indian Ocean, the SST peaks in April at about 29.5 °C and is at a minimum in December, at around 28 °C. In other words, the seasonal variation is about 1.5 °C, equal to the amplitude of the SST perturbation used in this study.

These SST perturbations are similar to, but not the same as, the SST biases in the coupled climate models in terms of the centre location, spatial extent, and magnitude. The ones used in EXP_POS and EXP_IOD have their counterparts in CMIP5 models (see Fig. 4 in Lyon, 2020). Those in EXP_NEG and EXP_10N are primarily used to study the sensitivity of the response to the sign and meridional location of the SST bias, respectively.

For each experiment, the model starts on 1 January 1901 and the simulation ends on 31 December 2010. The initial condition comes from a 40-year spin-up run (i.e. an equilibrium state) forced by the climatological monthly mean ERA-20C SST (without interannual variation). In the reference simulation, the heat fluxes at the Earth's surface and the top of the atmosphere are found to be balanced (i.e.

zero total heat flux) throughout the integration, and the total atmospheric energy (kinetic energy plus potential energy) is conserved, even during the initialization. When a strong SST bias is applied, it will lead to the breakdown in the model's surface–top CE2 heat flux balance during the first several months of the integration (not shown), which results in a net energy input (output) into (from) the atmosphere. After this short period, the surface–top heat flux balance recovers without further net energy input (output) into (from) the atmosphere, and the atmospheric total energy will remain at the level it attains. In this study, the SST bias applied is however too weak to affect the overall heat flux balance, and the total atmospheric energy is conserved over time. But note that the weak SST bias can result in notable modifications to the available potential energy and the kinetic energy, which are the primary focus of this study.

In addition, the atmospheric circulation bias can be established in the first decade, but the atmospheric variability takes several decades to achieve equilibrium. We therefore discard the first 30 integration years and use the last 80 years (1931–2010) in the analysis.

## 2.2 Modal analysis

The NMF decomposition of the PLASIM simulations is performed using the MODES software (Žagar et al., 2015). The global circulation is decomposed in terms of the Hough harmonics, which provide scale and regime information. Here we summarize the concept and refer the reader to Žagar et al. (2015) for details. Once the circulation is decomposed, statistics of biases and variances can be carried out in modal (or spectral) space as outlined below.

### 2.2.1 Circulation decomposition of balanced and unbalanced components

The scale and regime decomposition of the 3D global circulation relies on the representation of the global baroclinic atmosphere in terms of $M$ global shallow-water systems, each characterized by its own fluid depth for the horizontal motions that is known as the equivalent depth. The oscillations of the horizontal wind and geopotential height fields in various systems are coupled through equivalent depth, denoted $D$, with the vertical structure equation. Given a vertical mode $m$ ($m = 1, \ldots, M$), the horizontal motions are represented by a series of Hough harmonics, which are the product of the Hough vector functions in the meridional direction and plane waves in the zonal direction. The complex Hough expansion coefficient $\chi_n^k(m)$, which represents the two wind components $(u, v)$ and the geopotential height $h$, is obtained as

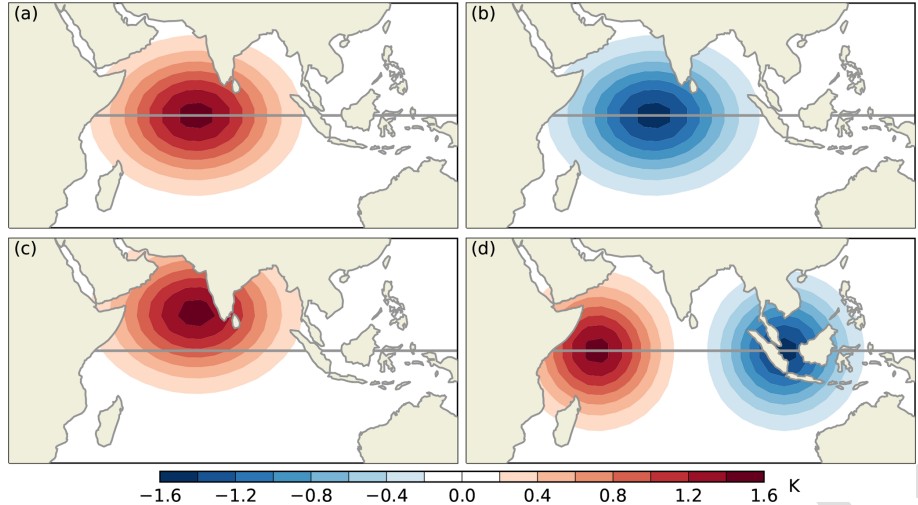

**Figure 1.** SST biases (in K) used in the sensitivity experiments: **(a)** EXP_POS, **(b)** EXP_NEG, **(c)** EXP_10N, and **(d)** EXP_IOD. The grey line marks the Equator.

$$\chi_n^k(m) = \frac{1}{2\pi} \int\limits_0^{2\pi} \int\limits_{-1}^{1} \left( \mathbf{S}_m^{-1} \sum_{j=1}^{J} (u, v, h)^T G_m(j) \right)$$
$$\cdot \left[ \boldsymbol{H}_n^k(m) \right]^* \mathrm{d}\mu \mathrm{d}\lambda, \tag{7}$$

where the asterisk ($*$) denotes the complex conjugate. Equation (7) describes a two-step procedure which consists of the vertical projection (within the parentheses) followed by the horizontal projection. The vertical structure functions $G_m(j)$ are orthogonal and solved using the finite difference method for the atmosphere discretized into $J$ $\sigma$ levels. The basis functions for the horizontal projection are the Hough harmonics, denoted by $\boldsymbol{H}_n^k(\lambda, \varphi; m)$. For any given vertical mode, the Hough harmonics are characterized by two indices: the zonal wavenumber $k$ and the meridional mode index $n$. The scaling matrix $\mathbf{S}_m$ is a $3 \times 3$ diagonal matrix which makes the input data after the vertical projection dimensionless. Parameters $\lambda$ and $\varphi$ stand for the geographical longitude and latitude, respectively, whereas $\mu = \sin(\varphi)$.

A discrete solution of Eq. (7) is obtained by replacing the integration with a finite series of the Hough harmonics, including the zonal-mean state, $K$ zonal waves, and $R$ meridional modes, which combine $N_R$ balanced or Rossby modes, $N_E$ eastward-propagating inertio-gravity (EIG) modes, and $N_W$ westward-propagating inertio-gravity (WIG) modes ($R = N_R + N_E + N_W$). Here we use a truncation similar to the PLASIM model, that is, $K = 30$, $M = 10$, and $N_R = N_E = N_W = 30$, meaning 30 meridional modes for each of the wave species. This means that our decomposition is complete and the statistics defined next account for the complete variance (or energy) in the system. The projection in this study is based on monthly mean data.

The decomposition (or filtering) of the circulation is an inverse process of the projection. One can filter any component of the circulation by selecting the corresponding Hough coefficients and reconstructing the component in physical space. In this way, the circulation can be decomposed into balanced and unbalanced components, with further decomposition into parts such as the Kelvin wave circulation, $n = 1$ Rossby wave circulation, and so on.

### 2.2.2 Statistics in modal space

The second moments of the circulation are evaluated in terms of the Hough coefficient $\chi_\nu(t)$, where a single modal index $\nu = (k, n, m)$. The total (kinetic plus available potential) energy, or the spatial variance, contained in mode $\nu$ per unit area at time step $t$ can be defined as (Žagar et al., 2020)

$$I_\nu(t) = \frac{1}{2} g D_m |\chi_\nu(t)|^2, \tag{8}$$

where $D_m$ is the equivalent depth of vertical mode $m$. The energy of the time-mean state (i.e. climatological energy) is

$$E_\nu = I(\overline{\chi_\nu}) = \frac{1}{2} g D_m |\overline{\chi_\nu}|^2, \quad \text{where } \overline{\chi_\nu} = \frac{1}{N} \sum_{t=1}^{N} \chi_\nu(t). \tag{9}$$

The temporal variance in modal space can be defined as

$$V_\nu = \frac{1}{N} \sum_{t=1}^{N} g D_m |\chi_\nu(t) - \overline{\chi_\nu}|^2. \tag{10}$$

As shown by Žagar et al. (2020), the global integration of $V_\nu$ is equivalent to the integral of the temporal variance in physical space:

$$\sum_k \sum_n \sum_m V_\nu =$$
$$\sum_i \sum_j w(\lambda_i, \varphi_j) \sum_{m=1}^M \left[ S_m^u + S_m^v + S_m^h \right]. \tag{11}$$

Here, $w(\lambda_i, \varphi_j)$ denotes the Gaussian weight, and

$$S_m^u = g D_m \mathrm{Var}(\tilde{u}_m),$$
$$S_m^v = g D_m \mathrm{Var}(\tilde{v}_m),$$
$$S_m^h = g D_m \mathrm{Var}(\tilde{h}_m), \tag{12}$$

where $\tilde{u}_m$, $\tilde{v}_m$, and $\tilde{h}_m$ represent the non-dimensionalized winds and height field after the vertical projection, and $\mathrm{Var}(x) = \frac{1}{N} \sum_{t=1}^{N} |x(t) - \overline{x}|^2$ denotes the temporal variance of any scalar variable $x$ at a location $(\lambda_i, \varphi_j)$.

Assuming ergodicity, the difference between the time-mean energy $\overline{I_\nu}$ and the energy of the mean state $E_\nu$ is equal to half the temporal variance (Žagar et al., 2020):

$$\overline{I_\nu} - E_\nu = \frac{1}{2} V_\nu. \tag{13}$$

Defining the time-mean energy of mode $\nu$ in the sensitivity simulation as $\overline{I_\nu^S}$ and that in the reference simulation as $\overline{I_\nu^R}$, we obtain

$$\overline{I_\nu^S} - \overline{I_\nu^R} = \left[ E_\nu^S - E_\nu^R \right] + \frac{1}{2} \left[ V_\nu^S - V_\nu^R \right], \text{ or }$$
$$\Delta \overline{I_\nu} = \Delta E_\nu + \frac{1}{2} \Delta V_\nu. \tag{14}$$

Superscripts S and R in Eq. (14) denote the sensitivity and reference simulations, respectively. It states that changes in the time-mean energy are attributed to changes in the climatological state $\Delta E_\nu$ and changes in the temporal variance $\Delta V_\nu$, allowing for an assessment of the variance budget associated with the SST bias. With the circulation bias in mode $\nu$ defined as

$$\overline{\Delta \chi_\nu} = \frac{1}{N} \sum_{t=1}^N \left[ \chi_\nu^S(t) - \chi_\nu^R(t) \right], \tag{15}$$

$\Delta E_\nu$ can be decomposed into two parts, $B_\nu$ and $P_\nu$, called the bias variance and covariance respectively, defined as

$$B_\nu = \frac{1}{2} g D_m \left| \overline{\Delta \chi_\nu} \right|^2 \text{ and}$$
$$P_\nu = \frac{1}{2} g D_m \left( \overline{\Delta \chi_\nu} \, \overline{\chi_\nu^R}^* + \overline{\Delta \chi_\nu}^* \, \overline{\chi_\nu^R} \right),$$
$$\text{with } \Delta E_\nu = B_\nu + P_\nu. \tag{16}$$

The bias covariance $P_\nu$ is computed between the bias $\overline{\Delta \chi_\nu}$ and the time-averaged reference state $\overline{\chi_\nu^R}$. These two terms indicate the amplitude ($B_\nu$) and phase ($P_\nu$) of the bias. To

see this, let $\overline{\chi_\nu^R} = \left| \overline{\chi_\nu^R} \right| e^{i\theta_A}$ and $\overline{\Delta \chi_\nu} = \left| \overline{\Delta \chi_\nu} \right| e^{i\theta_B}$, and then the bias covariance becomes

$$P_\nu = g D_m \left| \overline{\chi_\nu^R} \right| \left| \overline{\Delta \chi_\nu} \right| \cos(\Delta \theta), \tag{17}$$

where $\Delta \theta = \theta_B - \theta_A$ denotes the phase difference between the bias and reference state. If $|\Delta \theta| < \pi/2$, then $P > 0$; otherwise, $P \leq 0$. If the model is bias free (i.e. $\overline{\Delta \chi_\nu} = 0$), both terms vanish.

## 3 Results

In this study, we aim at understanding the general principle behind the bias teleconnection rather than its seasonal variation. Indeed, the circulation bias in the tropics (25° S–25° N) has a similar pattern throughout the year but with varying magnitude, and the extratropical biases are primarily observed in the Northern Hemisphere during boreal winter and in the Southern Hemisphere during boreal summer. In transition seasons (spring and autumn), extratropical biases exist in both Hemispheres with similar magnitude. We narrow our focus exclusively to the boreal wintertime (December–January–February; DJF). In the following, we first validate the reference simulation before discussing the bias and variability changes in sensitivity simulations.

### 3.1 Model validation

Figure 2 displays the climatology DJF fields of the reference simulation and the ERA-20C reanalyses. Overall, PLASIM is able to simulate the precipitation and the general circulation with correct patterns and magnitudes. For instance, the strong precipitation centres in the tropics are well simulated, which indicate the Inter-Tropical Convergence Zone (ITCZ) and the South Pacific Convergence Zone (SPCZ) (Fig. 2a). In midlatitudes, strong precipitation is seen in the North Pacific and North Atlantic, where the atmospheric storm tracks are located. As for the circulation, the high–low-pressure systems at the sea level (Fig. 2b) and zonal wind in the upper troposphere (Fig. 2c) are also well reproduced. However, there are also some discrepancies between the PLASIM simulation and the reanalyses. Particularly, the precipitation centres in the Bay of Bengal and the South China Sea are not well simulated (Fig. 2a). Besides, the precipitation in midlatitudes is generally underestimated, and the precipitation centre over the North Pacific shifts northward compared with the reanalysis, which should be attributed to the northward shift of the Aleutian low (Fig. 2b) and the jet stream (Fig. 2c) over the North Pacific in the simulation. Nevertheless, these discrepancies are not unexpected given that PLASIM is a model of intermediate complexity.

In what follows, we validate sensitivity simulations against the reference simulation and according to Eq. (15) refer to their time-averaged (1931–2010) difference as "bias".

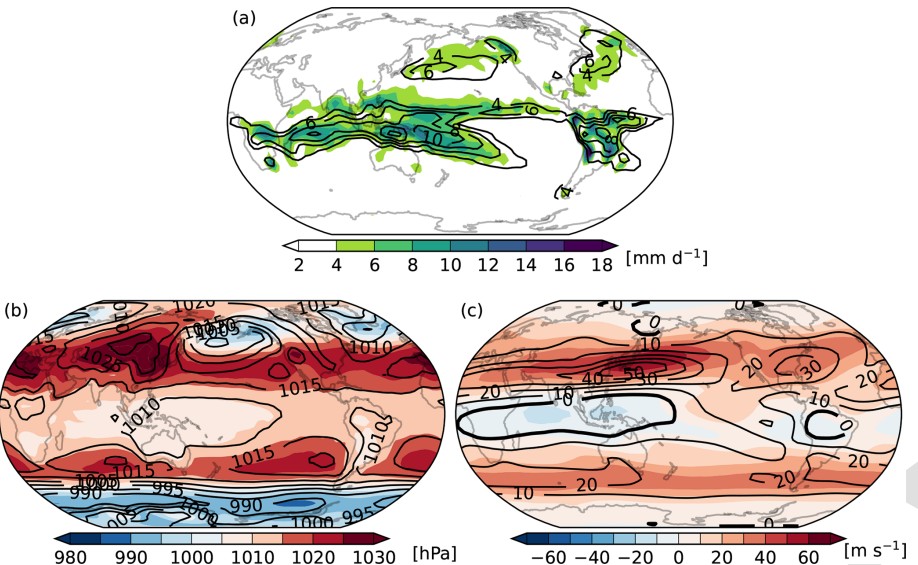

**Figure 2.** Long-term (1931–2010) mean DJF fields of the reference simulations (colours) overlaid with the ERA-20C reanalyses (contours): **(a)** total precipitation (in mm d$^{-1}$), **(b)** mean sea level pressure (in hPa), and **(c)** 250 hPa zonal wind (in m s$^{-1}$).

## 3.2 Precipitation biases

Figure 3 shows the distribution of the DJF total precipitation biases. The strongest biases are confined to the tropics. One prominent common feature among the four experiments is the wet (dry) bias in the area where positive (negative) SST biases are applied. As explained in Sect. 2.1.1, this is attributed to the response of the local air–sea interaction to the SST bias. The warm SST bias results in locally more upward, sensible and moisture fluxes at the surface, lowering the near-surface static stability, enhancing the low-level moisture convergence (Lindzen and Nigam, 1987; Back and Bretherton, 2009), and therefore bringing more deep convection and precipitation. The cold SST bias leads to opposite results, namely dry bias in the TIO region (Fig. 3b). The northward shift of the SST bias does not have much impact on the spatial structure of the precipitation biases but reduces the magnitude (Fig. 3c). As for the dipolar SST bias, it causes much stronger precipitation biases in the TIO region (Fig. 3d) than monopolar SST biases (Fig. 3a–c).

Apart from the local response, significant biases are also seen in remote areas. Looking at EXP_POS (Fig. 3a), strong dry biases are found roughly along the Equator to the west and east of the TIO region. In particular, the rain band associated with the SPCZ is significantly reduced. Significant precipitation biases are also visible in the extratropics (e.g. East and North Asia and North Pacific), but their magnitudes are small. These nonlocal precipitation biases are associated with the circulation biases caused by the SST biases. We will come back to this later.

## 3.3 Circulation biases

In this section, we discuss biases in circulation decomposed in balanced (Rossby, including the mixed Rossby-gravity (MRG) modes) and unbalanced (IG, including the Kelvin modes) components. But first we present the total circulation bias, that is, the sum of the balanced and unbalanced components.

### 3.3.1 Total biases

Biases in the DJF horizontal circulation are shown in Fig. 4. In all sensitivity experiments, strong biases are mainly observed in the tropics and the Northern Hemisphere. In the experiment with positive (negative) SST biases, anomalous winds associated with negative (positive) geopotential height biases converge (diverge) at lower levels in the TIO area, whereas at upper levels the winds diverge (converge); this indicates the Walker circulation bias on the zonal section, which is closely associated with the strong precipitation biases in the tropics (Fig. 3). In the extratropics, the bias centres at lower and upper levels are generally in phase, indicating barotropic structure. These centres are organized as stationary Rossby wave trains linking the subtropics and midlatitudes to high latitudes, which bear some resemblance to the simulation by Annamalai et al. (2007). They should account for the extratropical precipitation biases (Fig. 3). For instance, in EXP_POS, one can see a cyclonic circulation bias and an anticyclonic circulation bias over the North Pacific (Fig. 4b), accompanied by positive and negative precipitation biases, respectively (Fig. 3a). In comparison, EXP_POS (Fig. 4a and b) and EXP_NEG (Fig. 4c and d) have similar bias distributions but opposite signs. Despite the north-

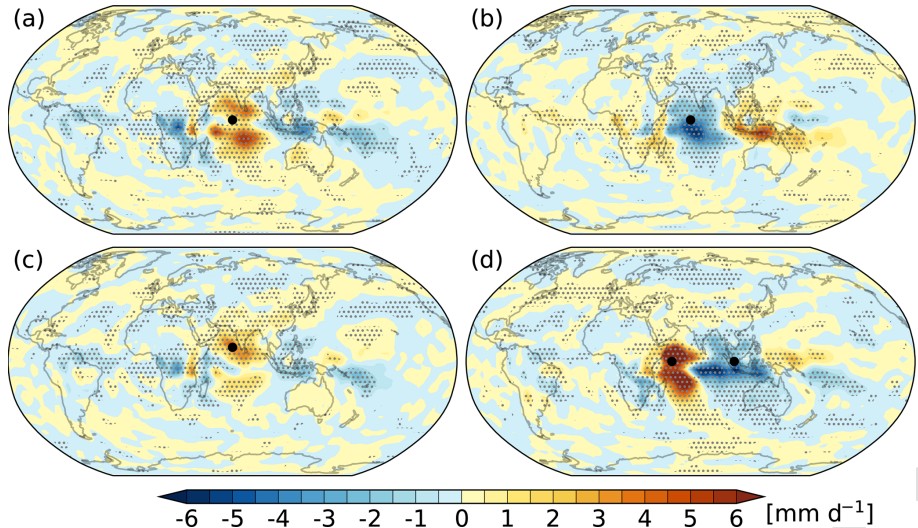

**Figure 3.** Distribution of the DJF precipitation biases (in mm d$^{-1}$): **(a)** EXP_POS, **(b)** EXP_NEG, **(c)** EXP_10N, and **(d)** EXP_IOD. The dotted area shows the region of biases statistically significant at 0.05 level using Student's *t* test. The large black dot in each panel marks the location of the SST bias centre (the same below).

ward shift of SST bias in EXP_10N, its circulation biases (Fig. 4e and f) resemble those in EXP_POS (Fig. 4a and b), albeit with smaller amplitude. Previous studies have shown that SST changes in tropical ascending regions are more efficient at generating global impacts (e.g. Zhou et al., 2017). In DJF, the ascending branch of the Hadley circulation is located slightly south of the Equator. Thus, as the SST bias moves northward away from the Equator, it becomes less efficient at impacting the atmosphere. The strongest extratropical bias centres are mainly observed over the Pacific–North America (PNA) region in all experiments except EXP_IOD, where they are seen over North America, North Atlantic, Europe, and Northeast Asia (Fig. 4g and h).

It is difficult to tell the changes in the zonal-mean flow from Fig. 4. In fact, the unbalanced zonal-mean flow is dominated by the Hadley circulation (Pikovnik et al., 2022). It weakens with positive SST bias (EXP_POS, EXP_10N, and EXP_IOD), whereas it strengthens with monopolar negative SST bias (EXP_NEG). But, in any case, the overall change is weak compared to the reference state. Regarding the balanced part, the diabatic heating induced by the positive SST bias warms the atmosphere and raises the geopotential field, particularly in the subtropics. This is accompanied by the strengthening of the westerlies in the midlatitudes and the easterlies in the tropics. The maximum change in the upper troposphere can reach $1.5\,\mathrm{m\,s^{-1}}$. The opposite occurs in monopolar negative SST bias forcing. The changes in the zonal-mean fields are further elaborated on in the Supplement.

In short, SST biases in the TIO region cause anomalous deep convection accompanied by anomalous diabatic heating, which ultimately leads to circulation biases worldwide. Nevertheless, the circulation biases are yet to be fully un-

derstood since the balanced and unbalanced components are mixed together, especially in the tropics. In the following, we decompose the total biases into the balanced and unbalanced regimes using the MODES software for further discussions.

### 3.3.2 Regime decomposition of the biases

We first look at the unbalanced biases, which are displayed in Fig. 5. They are reconstructed with the Hough coefficients of all unbalanced modes except the zonal-mean ($k = 0$) mode since we are more interested in the wave part. EXP_10N is not shown since it has quite similar results to EXP_POS (see Fig. 4). One common feature among these experiments is that the unbalanced biases are mainly confined to lower latitudes. They have the characteristics of the baroclinic Kelvin waves (Matsuno, 1966). The geopotential height field exhibits a dipolar structure along the Equator where the wind and mass fields are generally balanced in the meridional direction and unbalanced in the zonal direction. This feature is however not clearly seen in the total fields (Fig. 4). In EXP_POS, the winds converge towards the warm area along the Equator at the lower level (Fig. 5a), whereas at the upper level the winds flow zonally away from the area (Fig. 5b). EXP_NEG has similar bias distributions to EXP_POS, but with opposite signs and smaller magnitudes (Fig. 5c and d), implying the non-linearity of the response to positive and negative SST biases (e.g. Lunkeit and von Detten, 1997). In EXP_IOD (Fig. 5e and f), the Kelvin-type biases shift a bit westward compared to EXP_POS.

Figure 6 shows the balanced biases. They also share a number of common features among the four experiments. The bias fields are characterized by a quadrupole in the tropics and subtropics (referred to from now on as TROP). In

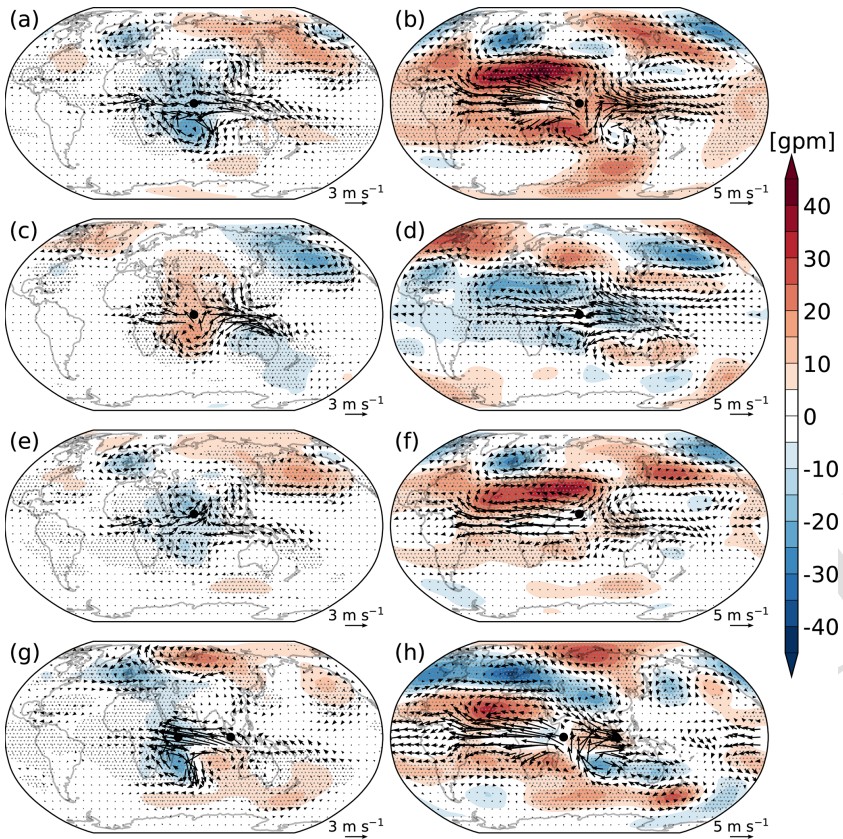

**Figure 4.** Total circulation biases in (**a** and **b**) EXP_POS, (**c** and **d**) EXP_NEG, (**e** and **f**) EXP_10N, and (**g** and **h**) EXP_IOD at (left) $\sigma = 0.924$ and (right) $\sigma = 0.211$. Vectors stand for winds (in m s$^{-1}$) and colours for geopotential height (in gpm). The dotted area shows the region of geopotential height biases statistically significant at 0.05 level using Student's $t$ test.

the extratropics (referred to from now on as EXTR), alternatively distributed cyclones and anticyclones organized in stationary Rossby-wave-train-like pattern are observed. Vertically, the TROP biases are baroclinic, similar to their unbalanced counterparts (Fig. 5). In contrast, the EXTR biases are generally barotropic. The biases in EXP_POS (Fig. 6a and b) and EXP_NEG (Fig. 6c and d) have similar distributions but opposite signs. In the extratropics, strong biases are particularly seen in the PNA region. Although EXP_IOD adopts a dipolar SST bias, its TROP bias fields are largely defined by the positive SST bias (Fig. 6e and f). The bias centres on EXP_IOD as a whole shift westward compared to the other experiments. This is consistent with previous studies that the atmospheric response to a dipolar heating in the tropics is mostly defined by the positive pole (Kosovelj et al., 2019). In contrast to the other experiments, the EXTR biases in EXP_IOD are strong over North America and Eurasia but relatively weak over North Pacific (Fig. 6f).

### 3.3.3 Mechanism of bias teleconnections

The TROP and EXTR balanced biases exhibit teleconnections, which can be demonstrated by using the stationary wave activity flux (WAF; Plumb, 1985) and the Rossby wave source (RWS; Sardeshmukh and Hoskins, 1988). The WAF is an indicator of the propagation of the Rossby wave activity, and the RWS denotes the wave forcing, which takes the form $-\nabla \cdot \mathbf{v}_\chi (\zeta + f)$, with $\mathbf{v}_\chi$ representing the divergent wind, $\zeta$ the relative vorticity, and $f$ the Coriolis parameter. Positive (negative) RWS indicates cyclonic (anticyclonic) wave forcing. The bias-related WAF and RWS are evaluated at 250 hPa. In the evaluation of WAF, the stationary wave fields are referred to as the balanced biases with the zonal-mean part removed. The RWS bias is computed as the time–mean difference between the sensitivity and the reference simulations.

The maps of RWS and WAF associated with balanced wave biases are displayed in Fig. 7. The major feature is the wave propagation indicated by WAF over Asia and the PNA region. In EXP_POS, there are two wave paths. One (the northern path) originates in Asia and spreads northeastward and then eastward; the other (the southern path) originates in the subtropical North Pacific and propagates northeastward. The two wave paths merge over the northeastern Pacific and then propagate eastward across the North America and North Atlantic, and finally they terminate over North

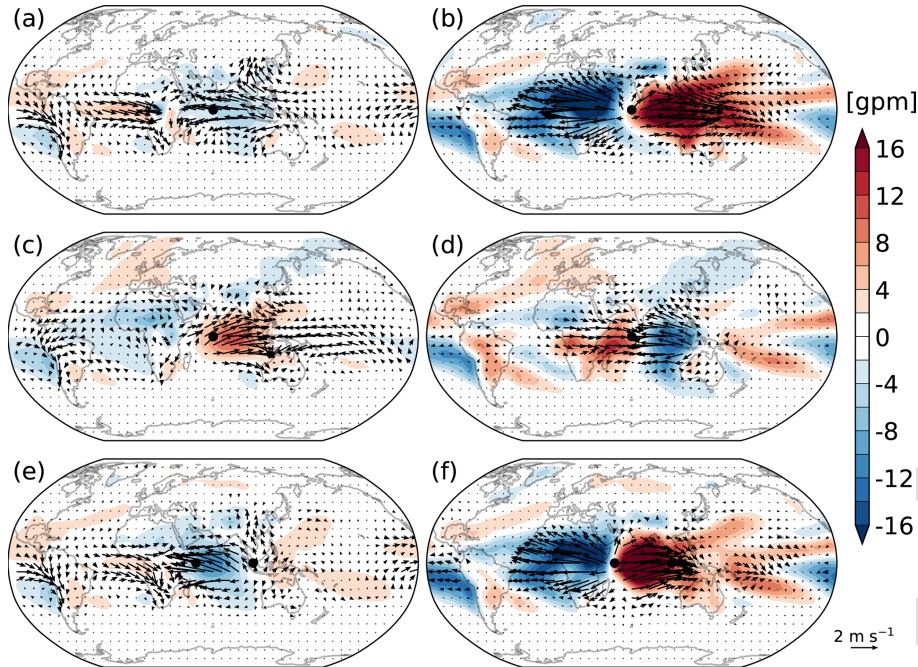

**Figure 5.** The unbalanced wave biases in (**a** and **b**) EXP_POS, (**c** and **d**) EXP_NEG, and (**e** and **f**) EXP_IOD at (left) $\sigma = 0.924$ and (right) $\sigma = 0.211$. Vectors stand for winds (in m s$^{-1}$) and colours for geopotential height (in gpm). The zonal-mean ($k = 0$) mode has been excluded.

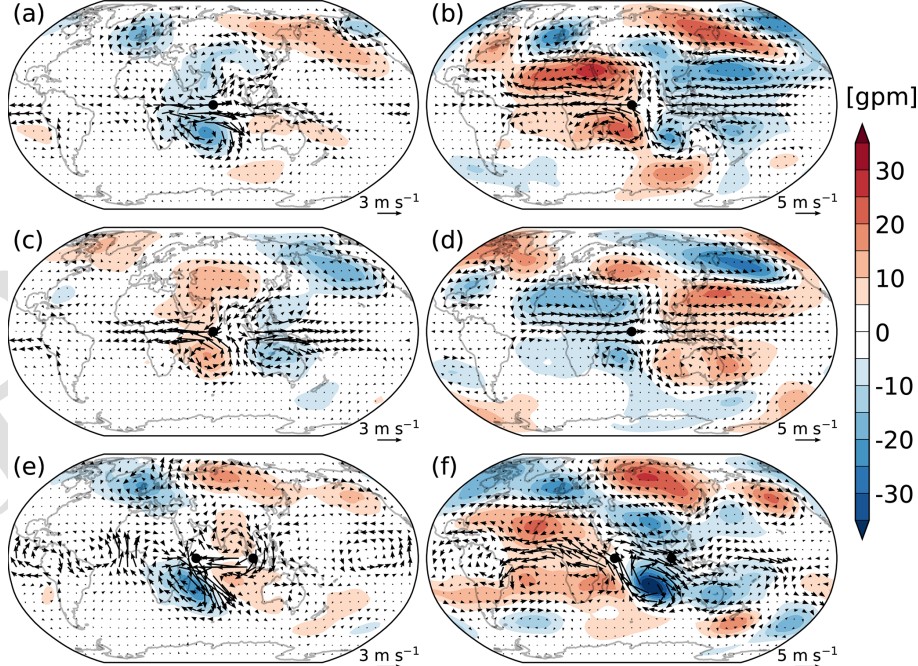

**Figure 6.** Same as Fig. 5 but for the balanced wave biases.

Africa (Fig. 7a). EXP_NEG has a similar wave propagation to EXP_POS, but its northern wave path is very weak (Fig. 7b). In EXP_10N, the northern wave path is similar to that of EXP_POS, but the southern wave path no longer exists (Fig. 7c). The wave train in EXP_IOD originates in South

Asia. It first spreads northeastward and then eastward across the North Pacific (Fig. 7d). The wave route in EXP_IOD is zonal to a large extent, which may be due to the wave being trapped by the jet stream (Zhang and Liang, 2022). The termination of the Rossby waves over America in EXP_IOD is

probably due to the zonal inhomogeneity of the jet stream, which is very weak to the west coast of North America and does not support Rossby wave propagation.

In addition, one can see that the wave propagation in all experiments originates from the subtropics where RWS is strong. The strong RWSs first generate the systems in the subtropics, which then disperse northeastward. For instance, over middle to East Asia, cyclonic (anticyclonic) circulations always correspond to some positive (negative) RWSs. These RWS centres are closely related to the subtropical jet (see Fig. 2c), where there is strong vorticity gradient, and the RWS can be effectively produced through the vorticity advection by divergent wind $[-\boldsymbol{v}_\chi \cdot \nabla(\zeta + f)]$. This is consistent with the conventional understanding of the teleconnection induced by tropical heating (e.g. Trenberth et al., 1998).

### 3.4 Bias and variance quantification

In this section, we look at the impact of the biases on the simulated spatio-temporal variability in both modal and physical spaces. As elucidated in Sect. 2.1.2, while the total energy (kinetic energy plus potential energy) remains conserved under weak SST biases, the available potential energy, as well as the kinetic energy, does not. In addition, the TROP and EXTR balanced biases, implied by their vertical structures, have different origins and dynamics and may have different influences on the simulated variability. Therefore, we first separate them before discussing the variability changes.

#### 3.4.1 TROP–EXTR separation of the balanced biases

The separation is performed in the modal space with the aid of the $Z$ profiles (the third component of the Hough vector). Although NMFs are global functions, they do have local features. Figure 8a displays some $Z$ profiles of the Rossby modes. One can see that their structures depend on the mode index $\nu = (k, n, m)$. In general, the $Z$ maxima (or minima) shift equatorward with $m$ and $k$ and move poleward with $n$. The latitude of $Z$ maximum (or minimum) indicates the action centre of the Rossby modes. In this study, we classify the Rossby mode that has its $Z$ maximum (or minimum) located between 25° S and 25° N as the TROP mode; otherwise we put it in the EXTR category. We also used latitude ±30° as the separation criterion and got the same results.

Figure 8b shows the separation in the meridional and vertical mode index plane at zonal wavenumber $k = 1$. We see that all modes with vertical index $m = 1$ or 2 (i.e. the tropospheric barotropic modes) are identified as the EXTR modes, consistent with the observations that the extratropical Rossby waves are generally barotropic (see Fig. 6). But note that the extratropical biases are not purely barotropic. As the meridional index $n$ increases, baroclinic modes go into the EXTR regime. Also note that the structure of $Z$ depends on $k$ as well, but it changes slowly with $k$. Indeed, the separations are almost identical among zonal wavenumbers $k < 7$ (not

shown). Finally, 1337 TROP modes and 6493 EXTR modes are identified (the zonal-mean modes and the MRG modes are excluded in the classification). The TROP and EXTR balanced biases are then reconstructed with the Hough coefficients of the corresponding categories.

An example of the reconstructed TROP and EXTR biases from EXP_POS is shown in Fig. 8c–f. As expected, the TROP biases are confined to the tropics, which are characterized by baroclinic equatorial Rossby waves (Fig. 8c and d). The EXTR biases are dominated by midlatitude to high-latitude barotropic systems, with tropical signals being very weak (Fig. 8e and f). Nevertheless, there is no clear geographical boundary between the TROP and EXTR biases. The TROP–EXTR separation can benefit us in understanding both local and remote impacts of these biases on the spatio-temporal variability.

#### 3.4.2 Variance budget in modal space

Let us first discuss the overall budget (Eq. 14). Figure 9 shows the changes of the time-mean energy ($\Delta \bar{I}$), the climatological energy ($\Delta E$), and the temporal variance ($\Delta V$). Since the DJF-mean fields are used in the computation, the temporal variance $V$ indicates the interannual variability. These quantities are calculated separately for the zonal-mean ($k = 0$) and the wave ($k > 0$) modes. The TROP and EXTR balanced wave modes are also treated separately. A positive (negative) value means an increase (decrease) of energy or variance with respect to the reference simulation.

One can find that $\bar{I}$ and $E$ have similar changes regardless of regimes, implying that $E$ dominates $V$ in the $\bar{I}$ budget (Eq. 14). For the zonal-mean ($k = 0$) modes, they decrease in the unbalanced part (Fig. 9a), whereas they increase in the balanced part (Fig. 9b) in all experiments. The overall changes vary with experiments, but their magnitudes are generally smaller than 5 % of the reference. The decrease in the unbalanced zonal-mean energy can be attributed to the weakening of the Hadley circulation, whereas the increase in the balanced zonal-mean energy is due to the diabatic heating induced by a positive SST bias and the strengthening of westerlies in midlatitudes and easterlies in the tropics (see the Supplement).

For the wave ($k > 0$) modes, the unbalanced $\bar{I}$ and $E$ increase in all experiments except EXP_IOD (Fig. 9c), similar to that of the TROP balanced modes (Fig. 9d). It is not surprising to see the similarity between the changes of these two wave species. Because the Kelvin modes dominating the unbalanced flow and the equatorial Rossby modes dominating the TROP balanced flow form the overall response to a tropical heating (Matsuno, 1966; Gill, 1980), they change in phase. Regarding the EXTR balanced modes, $\bar{I}$ and $E$ decrease in all experiments except for the rise in EXP_NEG (Fig. 9e). This implies a different scenario of the nonlinear interaction between the wave flow and the zonal-mean flow in EXP_NEG from the others. The wave circulation biases

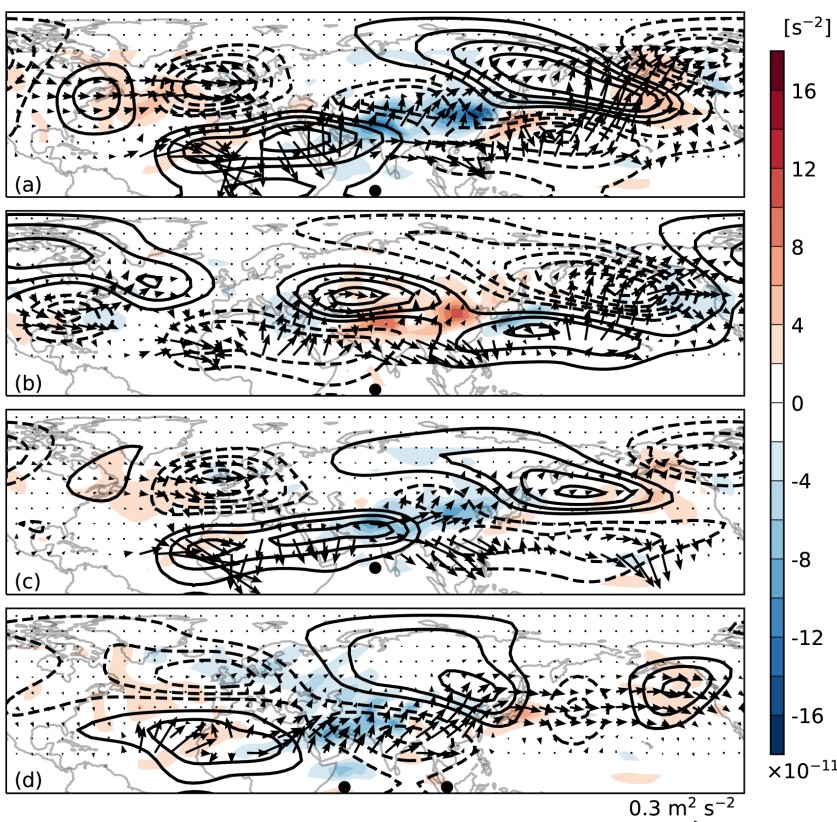

**Figure 7.** Horizontal distribution of the 250 hPa stationary wave activity flux (vectors, $m^2\,s^{-2}$) and Rossby wave source (colours, $s^{-2}$) in DJF: **(a)** EXP_POS, **(b)** EXP_NEG, **(c)** EXP_10N, and **(d)** EXP_IOD. Black contours show the balanced geopotential height biases. The contour interval is 5 gpm. Negative values are indicated with dashed lines and the zero line is omitted.

caused by SST biases can modify the zonal-mean flow, which in turn alters the nonlinear interaction between them. It is found that the biases in EXP_NEG act to strengthen the baroclinic energy transfer from zonal-mean flow to wave flow, leading to a wave energy increase (not shown). The magnitudes of the relative changes in the unbalanced wave energy are generally smaller than 5 % (Fig. 9c), and those in the TROP balanced wave energy range from 3 % (EXP_10N) to 15 % (EXP_NEG) (Fig. 9d). The relative changes of the EXTR balanced wave energy stay around 5 % in all experiments (Fig. 9e).

The energy changes can be further understood through $B$ and $P$ terms, the two components of $\Delta E$ (Eq. 16). One can see that the $P$ term dominates the $B$ term in the zonal-mean modes (Fig. 9a and b) as well as the EXTR balanced wave modes (Fig. 9e). In fact, the unbalanced (balanced) zonal-mean biases are out of (in) phase with the reference state (see Supplement), resulting in large negative (positive) covariance between them. This indicates the importance of the bias phase to energy changes in the zonal-mean and the EXTR balanced wave modes. Contrarily, the $B$ term dominates the $P$ term in the unbalanced wave modes (Fig. 9c) and

the TROP balanced wave modes (Fig. 9d) except EXP_IOD, where the opposite situation happens.

Regarding $V$, it decreases in the unbalanced zonal-mean $(k = 0)$ modes in all experiments (Fig. 9a). For the balanced zonal-mean $(k = 0)$ modes, $V$ increases in EXP_NEG and EXP_IOD and decreases in the other two experiments (Fig. 9b). For wave modes of all kinds, $V$ decreases in all experiments except EXP_NEG, where it increases (Fig. 9c–e), implying that positive (negative) TIO SST bias weakens (strengthens) the global interannual variability of the wave flow. Although $\Delta V$ in the zonal-mean flow is very small compared to $\Delta \overline{I}$ and $\Delta E$, the relative change $\Delta V / V$ could be over 10 % (Fig. 9a and b). A similar situation happens in the TROP wave modes (Fig. 9c and d). Comparisons between the tropical and extratropical budgets indicate that the TIO SST bias has relatively stronger influence on $V$ in the tropics (Fig. 9c and d) than in the extratropics (Fig. 9e) in terms of their relative changes.

The climatological energy change $\Delta E$ with respect to the zonal wavenumber $k$ is displayed in Fig. 10, which is similar to $\Delta \overline{I}$ (not shown). For the unbalanced regime, $\Delta E$ is dominated by zonal wavenumber $k = 1$ in EXP_POS and EXP_10N (Fig. 10a), largely due to the $B$ term (Fig. 10b).

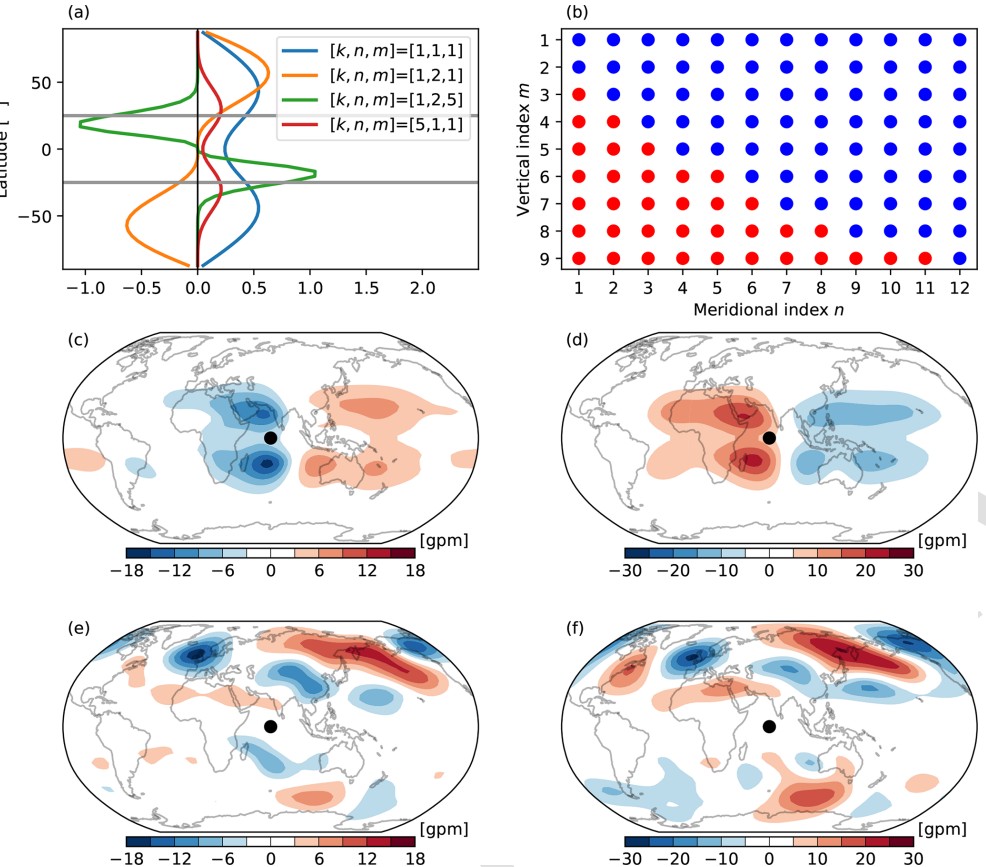

**Figure 8. (a)** Examples of the $Z$ profile of the Rossby mode. Grey lines mark the latitudes $\pm25°$. **(b)** Classification of the TROP and EXTR modes in modal space at zonal wavenumber $k = 1$ based on $Z$ profiles. Red (blue) dots denote the TROP (EXTR) modes. **(c and d)** Reconstructions of the TROP balanced geopotential height biases (m) in EXP_POS at **(c)** $\sigma = 0.924$ and **(d)** $\sigma = 0.211$. Panels **(e)** and **(f)** are the same as **(c)** and **(d)** but for the EXTR biases.

In EXP_NEG, $\Delta E$ is dominated by $k = 1$ and 2, which have comparable $B$ and $P$ terms (Fig. 10b and c). $\Delta E$ in EXP_IOD is dominated by $k = 1$–3 and is mostly attributed to the $P$ term, which has relatively large negative values at $k = 1$ and 2 (Fig. 10c). This could be related to the large phase differences (greater than $\pi/2$) between the climatological Walker circulation and the biases caused by dipolar SST bias (see the Supplement). Regarding the balanced regime, the TROP component has similar $k$ spectra to the unbalanced regime in terms of $\Delta E$, $B$, and $P$ (Fig. 10d–f), which has been explained in the previous sections. As for the EXTR component, $\Delta E$ is dominated by $k = 1$ in all experiments (Fig. 10g), predominately due to the $P$ term (Fig. 10i).

Another point worth noting is the spectra of the $B$ term, which indicates the bias amplitude (Fig. 10b, e, and h). They seem to decay exponentially with $k$. Quantitative calculations show that 90 %–95 % of the bias variance is stored in the first six zonal modes ($k = 0$–5; see Table 2), consistent with Žagar et al. (2020) that the representation of the SST is crucial for large scales. Besides, a large part (ranging from 73 % to 88 %) of the bias variance comes from the merid-

ional symmetric modes, which in the unbalanced regime are dominated by the Kelvin modes, and the MRG modes contribute less than 1 % to the balanced bias variance. This is even true in EXP_10N, where the erroneous SST forcing is situated north of the Equator. Furthermore, more bias variance comes from the baroclinic modes ($m = 3$ and 4) than the barotropic modes ($m = 1$ and 2). On the other hand, different SST biases result in different amounts of bias variance. For instance, EXP_POS and EXP_IOD have similar amounts of bias variance, stronger than the other two experiments.

The interannual variance change $\Delta V$ with respect to zonal wavenumber $k$ is displayed in Fig. 11. In contrast to $\Delta E$, more zonal modes engage in $\Delta V$. The unbalanced (Fig. 11a) and the TROP balanced (Fig. 11b) regimes have similar spectrum structures, albeit with different magnitudes. The zonal wavenumbers $k = 1$ and 2 dominate in EXP_POS and EXP_10N, and $k = 1$–4 dominate in EXP_NEG. A difference is seen in EXP_IOD, where $\Delta V$ is dominated by $k = 2$ and 3 in the unbalanced regime, whereas it is dominated by $k = 1$ and 2 in the TROP balanced regime. The $\Delta V$ spectra in the EXTR balanced regime differ (Fig. 11c). One can see that

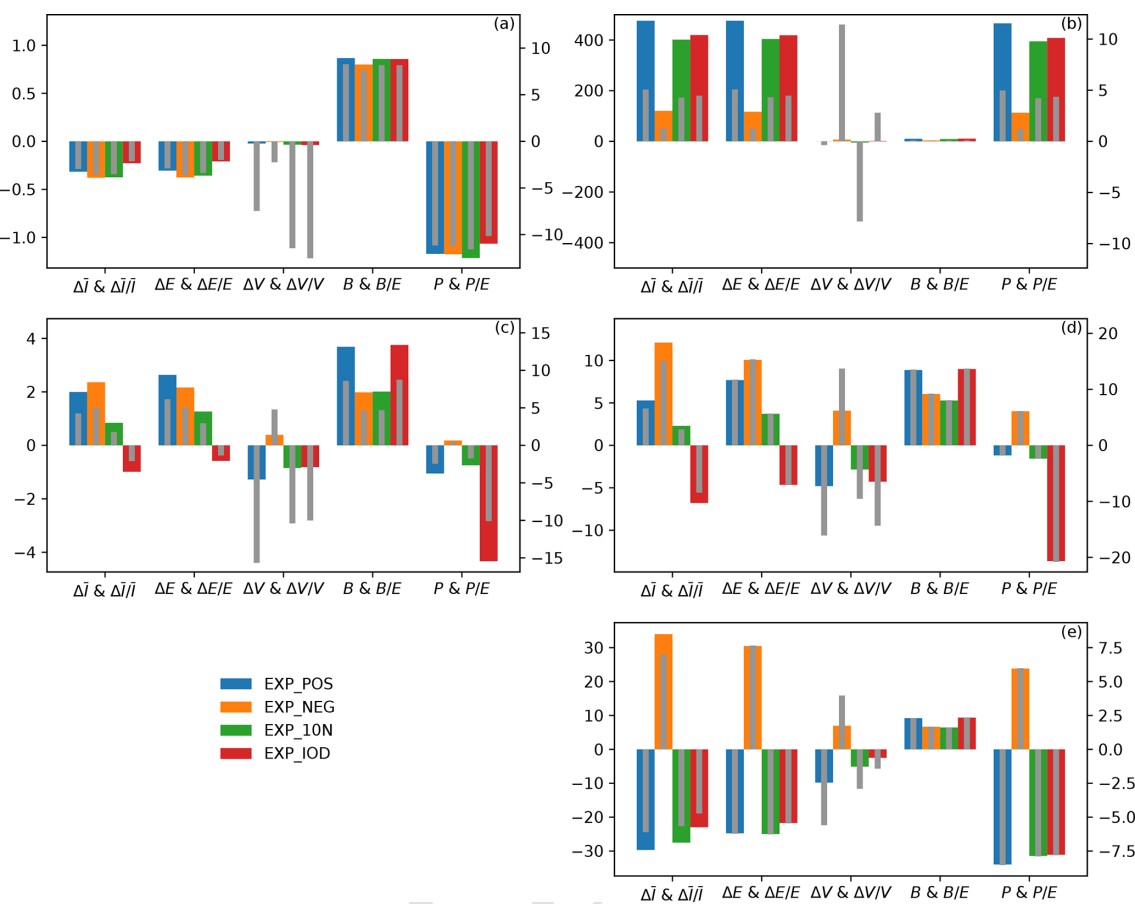

**Figure 9.** Absolute (wide bars) and relative (narrow bars) changes of the energy ($\overline{I}$ and $E$) and the interannual variance ($V$). $B$ and $P$ are the two components of $\Delta E$ (see Eq. 16). The left panels show the results of the unbalanced **(a)** zonal-mean ($k = 0$) modes and **(c)** wave ($k > 0$) modes. The right panels show the results of **(b)** the balanced zonal-mean ($k = 0$) modes, **(d)** the TROP balanced wave ($k > 0$) modes, and **(e)** the EXTR balanced wave ($k > 0$) modes. The relative change is calculated as the absolute change divided by the respective reference state. The left ordinate indicates the absolute changes (in $\mathrm{m^2\,s^{-2}}$), whereas the right axis indicates the relative changes (in %).

**Table 2.** Bias variance and the percentages of selected modes. The percentage is calculated as the bias variance of the selected modes divided by the respective total amount (including $k = 0$).

| Experiments | EXP_POS | | EXP_NEG | | EXP_10N | | EXP_IOD | |
|---|---|---|---|---|---|---|---|---|
| Regimes | Unbalanced | Balanced | Unbalanced | Balanced | Unbalanced | Balanced | Unbalanced | Balanced |
| Total variance ($\mathrm{J\,kg^{-1}}$) | 4.6 | 28.2 | 2.8 | 16.6 | 2.9 | 20.8 | 4.6 | 28.9 |
| $k = 0$ (%) | 19.0 | 35.9 | 28.7 | 23.4 | 29.9 | 43.5 | 18.6 | 36.5 |
| $k = 1$–5 (%) | 73.3 | 59.5 | 65.2 | 72.3 | 61.6 | 52.7 | 74.6 | 57.1 |
| $m = 1$–2 (%) | 10.9 | 24.8 | 16.6 | 28.0 | 16.5 | 28.5 | 11.0 | 29.3 |
| $m = 3$–4 (%) | 65.5 | 49.1 | 53.2 | 44.8 | 59.3 | 40.8 | 70.9 | 47.0 |
| Symmetric modes (%) | 88.1 | 77.1 | 83.2 | 73.2 | 83.0 | 73.6 | 87.9 | 73.6 |
| Kelvin (%) | 64.7 | – | 51.7 | – | 49.7 | – | 65.7 | – |
| MRG (%) | – | 0.8 | – | 0.91 | – | 0.7 | – | 1.0 |

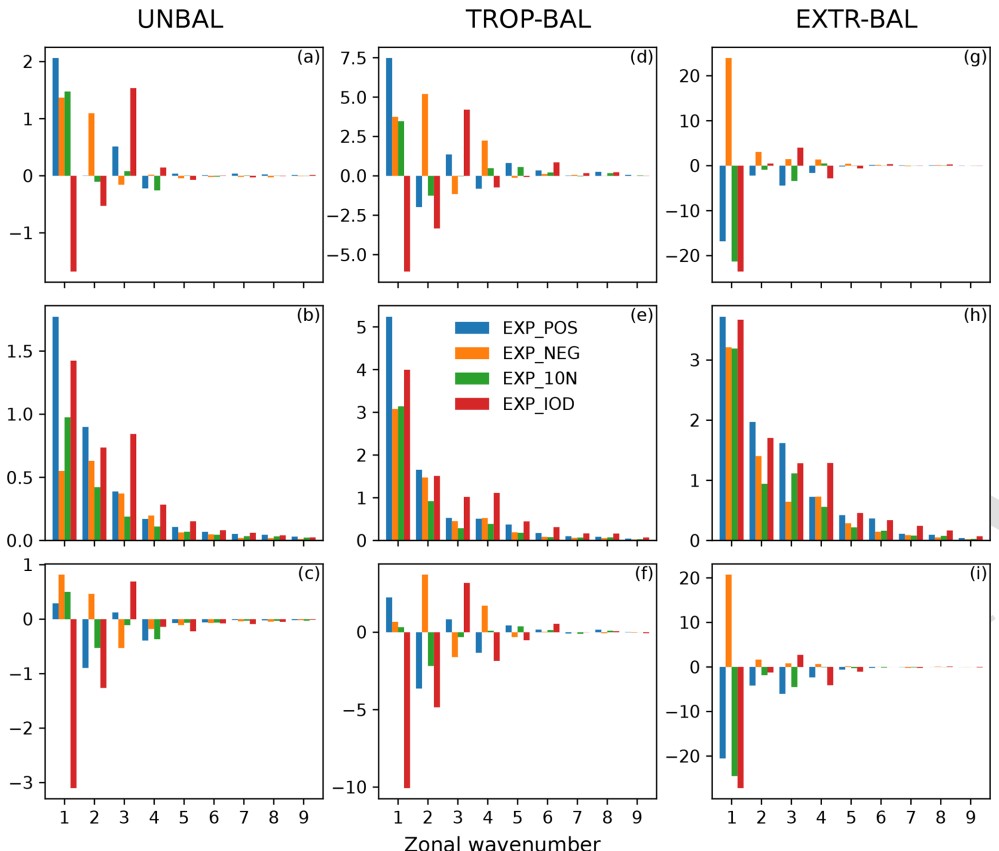

**Figure 10.** (left column) The spectra (in m$^2$ s$^{-2}$) of **(a)** $\Delta E$, **(b)** $B$, and **(c)** $P$ as a function of zonal wavenumber $k$ for the unbalanced modes, which have been summed over the meridional index $n$ and the vertical index $m$. Middle column: the same as left panels but for the TROP balanced modes. Right column: the same as left panels but for the EXTR balanced modes.

$\Delta V$ is dominated by $k = 1$–4 in EXP_POS, by $k = 1$–5 in EXP_NEG, and by $k = 2$ and 4 in EXP_10N. In EXP_IOD, $k = 1$, 3, and 4 dominate $\Delta V$ but with different signs; $k = 1$ is positive, whereas the other two are negative, which define the total change.

### 3.4.3 Interannual variance changes in physical space

The horizontal distribution of the interannual variance (IAV) changes is displayed in Fig. 12. The IAV is computed based on the fields after the vertical projection (Eq. 12). EXP_10N has similar distributions to EXP_POS and is thus not shown. In general, strong IAV changes are seen in regions where the background IAVs are strong, and the distributions of the IAV changes do not have big differences among experiments.

In the unbalanced regime, zonal winds (Fig. 12a–c) and geopotential height (not shown) dominate the IAV changes, which have similar structures and magnitudes. We see that large IAV changes are confined to low latitudes, especially the Indo-West Pacific region, the equatorial Atlantic and Africa, and South Asia and Australia, where they can be over 25 % of the reference.

In the balanced regime, large IAV changes are seen globally (Fig. 12d–l). At low latitudes, both zonal wind and geopotential height undergo large IAV changes, but their distributions are different. The IAV changes in the zonal wind are mainly distributed along the Equator in the Indo-West Pacific region (Fig. 12d–f), whereas those in the geopotential height are seen in subtropics, such as the Middle East, South Asia, Australia, and the subtropical central Pacific (Fig. 12j–l). In contrast, the absolute IAV changes in the meridional wind are weak in the tropics, but the relative changes can be very large (over 50 %), especially in the TIO region and the Maritime Continent (Fig. 12g–i). At midlatitudes to high latitudes, strong IAV changes generally occur in the PNA sector, North Atlantic, and Europe, although the locations of the change centres vary with variables. The IAV changes of the zonal wind are mostly seen in midlatitudes, especially over the North Pacific and western Europe (Fig. 12d–f). Those of the meridional wind (Fig. 12g–i) and geopotential height (Fig. 12j–l) are confined to high latitudes, especially over Alaska, North America, North Europe, and Northeast Asia. The IAV changes in the extratropics should be attributed to the impact of the EXTR balanced biases on wave and zonal-

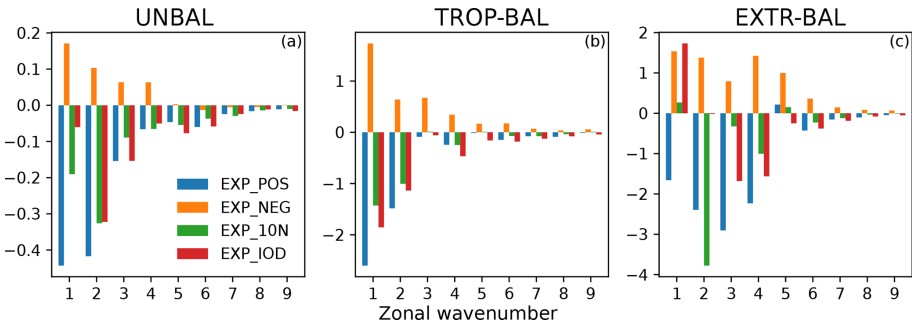

**Figure 11.** The spectra (in $m^2\,s^{-2}$) of $\Delta V$ as a function of zonal wavenumber $k$ for **(a)** the unbalanced modes, **(b)** the TROP balanced modes, and **(c)** the EXTR balanced modes. The results have been summed over the meridional index $n$ and the vertical index $m$.

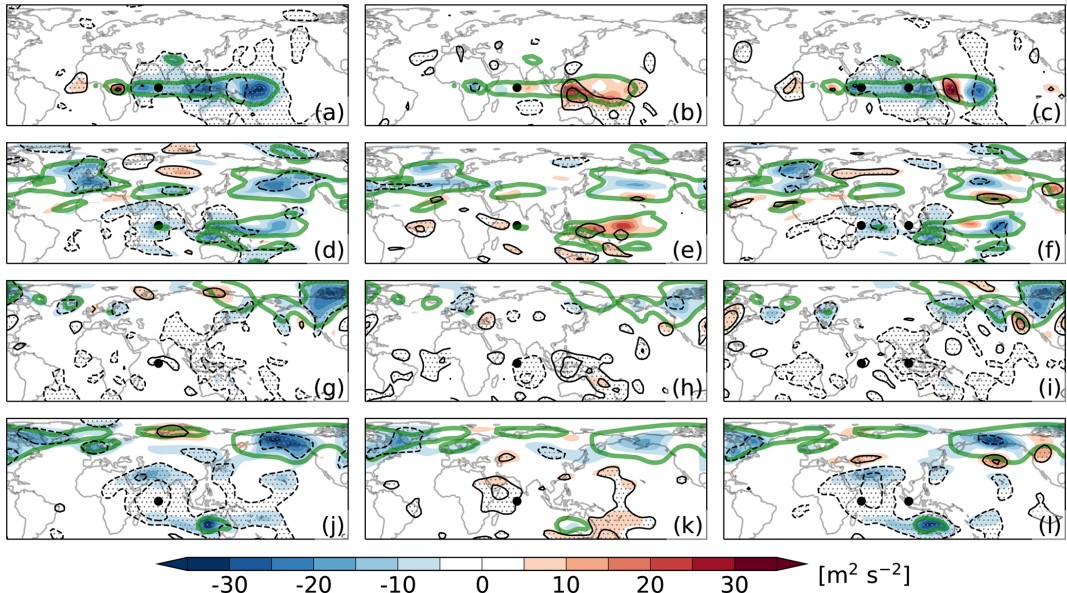

**Figure 12.** Absolute changes (colours; in $m^2\,s^{-2}$) and relative changes (black contours; in %) of the interannual variance (summed over the vertical index $m$, i.e. $\sum_m S_m$) with respect to the reference simulation: **(a–c)** unbalanced zonal wind, **(d–f)** balanced zonal wind, **(g–i)** balanced meridional wind, and **(j–l)** balanced geopotential height. The left column shows the results from EXP_POS, middle column from EXP_NEG, and right column from EXP_IOD. The zonal-mean modes are excluded in the computation. The absolute changes in panels **(a)**–**(c)** have been multiplied by 10 to facilitate drawing. Black contours are drawn at $\pm25\,\%$ and $\pm50\,\%$; negative values are indicated by dashed lines. Thick green contours show the respective reference variance at **(a–c)** 3, **(d–f)** 30, **(g–i)** 30, and **(j–l)** $40\,m^2\,s^{-2}$, respectively.

mean flow interaction (see Fig. 7). The EXTR balanced biases (i.e. stationary Rossby waves) could modify the background flow (e.g. jet stream), which in turn affects the nonlinear interaction between them, leading to changes in both energy and interannual variability (e.g. Lau and Nath, 1991; Wettstein and Wallace, 2010; Zhao and Liang, 2018).

## 4 Conclusions and outlook

This study investigated both the local and remote effects of regional SST biases in the tropical Indian Ocean (TIO) based on century-long (1901–2010) simulations by PLASIM, a GCM of intermediate complexity. Motivated by results from CMIP5 and CMIP6 simulations, the SST biases were prescribed as monopole or dipole anomalies with maximal amplitudes of $\pm1.5\,K$ in the TIO region, superimposed on the monthly SST from ERA-20C reanalyses. The effects of the TIO SST biases enter the atmosphere through moisture and heat fluxes at the surface, thereby affecting local precipitation and generating local and remote changes in the circulation. These effects are termed "bias teleconnections". For simplicity, we investigated the bias teleconnections with a focus on boreal wintertime (December–January–February; DJF). While simplified with respect to the real atmosphere and coupled climate models, the application of PLASIM with prescribed SST provides a framework for understanding the

changes in spatio-temporal variability associated with systematical errors in regional SST.

The results show that the bias teleconnections caused by the TIO SST biases are similar to the steady-state response to tropical diabatic heating. In general, the TIO SST biases induce circulation biases that have the structure resembling the Gill–Matsuno pattern in the tropics and Rossby-wave-train distribution in the extratropics, especially over the Pacific–North American (PNA) sector. The teleconnection between the tropical and extratropical biases is set up by Rossby wave activity flux emanating from the subtropical areas with strong Rossby wave sources. The biases mainly reside at large scales, with over 90 % of the bias variance (or squared amplitude) in zonal wavenumbers $k < 6$. Comparisons among experiments show that the northward shift of the SST bias away from the Equator weakens the atmospheric response, but it does not change its overall structure. Besides, the positive SST bias produces stronger bias teleconnections than the negative one of the same size and magnitude.

Our methodology also provides scale- and regime-decomposed biases in the atmospheric energy and interannual variance (IAV). The energy bias of the time-mean state can be represented as the sum of the bias variance and bias covariance, which measure the amplitude and phase, respectively, of the circulation bias. In addition, the energy bias is shown to be the difference between the bias in the time-mean energy (or spatial variance) and half the bias in the IAV (or transient energy).

The changes in atmospheric energy (i.e. spatial variance) induced by the TIO SST biases are analysed separately for the zonal-mean ($k = 0$) part and the wave ($k > 0$) part of the balanced and unbalanced components of the circulation. Across all experiments, the balanced zonal-mean flow energy increases by up to 5 % compared to the reference state, whereas the unbalanced part (only 0.1 % of its balanced counterpart) decreases by up to 3 %. These changes primarily arise from the covariance between the circulation bias and the reference state, which results in a weakening of the Hadley circulation in the unbalanced regime and a strengthening of the zonal-mean westerlies (easterlies) in the midlatitudes (tropics) in the balanced regime.

The wave energy response to the TIO SST bias is decomposed into three components: the unbalanced part, which is confined to the tropics, and the tropical (25° S–25° N) and extratropical balanced parts. The tropical wave energies in unbalanced and balanced regimes exhibit an in-phase response. They increase in experiments with a monopolar SST bias and decrease in the case with a dipolar SST bias. The increase is mainly due to bias variance, while the decrease is due to a strong negative covariance between the bias in unbalanced circulation and the reference state (characterized by Walker circulation) at zonal wavenumbers 1 and 2. In contrast, changes in the extratropical balanced wave energy depend on the sign of the SST bias. The positive SST bias (including the dipolar case) leads to an energy decrease, while the negative SST bias results in an energy increase. The primary factor contributing to the changes (decrease or increase) is the bias covariance at zonal wavenumber 1.

The IAV responses are contingent upon the sign of the SST bias. We found that the positive SST bias can lead to reductions in the IAV of up to 15 % for the unbalanced wave flow, 16 % for the tropical balanced wave flow, and 6 % for the extratropical balanced wave flow, respectively, compared to their respective reference states. The negative SST bias has the opposite effect, causing an increase in the IAV by up to 5 % for the unbalanced wave flow, 14 % for the tropical balanced wave flow, and 4 % for the extratropical balanced wave flow, respectively. In contrast to the energy response, changes in the IAV due to the TIO SST biases are spread across more zonal scales, with visible changes not only at planetary scales but also at zonal wavenumbers 4 and 5. Geographically, the IAV responses to the TIO SST bias are predominantly confined to the Indo-West Pacific region, Australia, South and Northeast Asia, the Pacific–North America region, and Europe, where the background IAVs are strong.

In summary, we conducted a thorough investigation of the bias teleconnections induced by SST biases in the TIO region and their impacts on model circulation and variability. Our paper provides a novel dynamical framework for evaluating simulations of CGCMs, such as CMIP models, that have suffered from severe SST biases over time. In the case of TIO SST biases, only positive SST biases are detrimental for the global zonal-mean circulation. In contrast, the responses of the wave circulation and its IAV depend on the sign of the SST bias. The IAVs of wave flows are reduced (enhanced) by the positive (negative) SST bias. In addition, the responses of the tropical wave energies are related to the spatial structure of the SST bias, with the monopolar (dipolar) SST bias increasing (decreasing) the tropical wave energies.

Finally, the simplifications of this study need to be mentioned. Using an atmosphere-only GCM forced with prescribed SST eliminates any feedback from the atmosphere to the ocean. While this simplifies the problem under study, it reduces the reality of simulated processes in the TIO region. In fact, the SST variability in the region is coherent with monsoon variability with a phase relation consistent with a coupled oscillation (Vecchi and Harrison, 2002), which is absent from the atmosphere-only GCM. The coupling between the SST and the precipitation (deep convection) could therefore be more complicated than depicted in this study. Furthermore, the TIO has warmed faster than any other tropical oceans over the past century (Roxy et al., 2020), and the atmospheric bias teleconnections associated with the TIO SST bias may have a temporal evolution with the non-stationary background SST. While not addressed in this paper, it will be discussed in follow-on studies. On the other hand, SST biases in different regions are likely to result in different bias teleconnections (e.g. Barsugli and Sardeshmukh, 2002; Zhou et al., 2017; Thomson and Vallis, 2018). Our future work will extend the approach developed in this study to other regions

of the global oceans to quantify the sensitivity of the atmospheric bias teleconnections to the location of the SST bias.

*Code and data availability.* All data used in this work can be obtained from Yuan-Bing Zhao through email. The PLASIM source code and detailed user guide can be downloaded at https://www.mi.uni-hamburg.de/en/arbeitsgruppen/theoretische-meteorologie/modelle/plasim.html (Universität Hamburg, 2023a). The MODES package can be requested via https://modes.cen.uni-hamburg.de (Universität Hamburg, 2023b).

*Supplement.* The supplement related to this article is available online at: https://doi.org/10.5194/wcd-5-1-2023-supplement.

*Author contributions.* YBZ and NŽ designed the study and YBZ carried them out. All co-authors contributed to interpreting the results. YBZ prepared the manuscript with contributions from all co-authors.

*Competing interests.* The contact author has declared that none of the authors has any competing interests.

*Acknowledgements.* This work is a contribution to project S1 of the Collaborative Research Centre TRR 181 "Energy Transfer in Atmosphere and Ocean". TS1

*Financial support.* This research has been supported by the Deutsche Forschungsgemeinschaft (grant no. 274762653).

*Review statement.* This paper was edited by Tim Woollings and reviewed by Ronald Kwan Kit Li and one anonymous referee.

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

**Remarks from the language copy-editor**

CE1     Please double-check the placement of "otherwise" now. It was adjusted.

CE2     The only adjustment necessary here is from a hyphen to an en dash. It is otherwise clear.

**Remarks from the typesetter**

TS1     Please provide a new text for this section if you would like to thank the National Natural Science Foundation of China. Thank you.

TS2     Please provide last access date.

TS3     Please check URL and provide last access date.