# Peer review of "Atmospheric bias teleconnections in boreal winter associated with systematic SST errors in the tropical Indian Ocean"

_EGUsphere, 2023_

## Author Comment (AC3)

Reviewer 1

Dear referee,

Thanks a lot for your thorough and insightful review. The following is a point-to-point reply. The answers are in blue.

General comments:

This study investigates how the tropical Indian Ocean SST biases that are typically seen in climate models may affect atmospheric teleconnections. To simplify the complex atmosphere-ocean coupling situation, this study makes use of a simpler atmospheric model framework where the SST biases are prescribed to force the atmospheric circulation. There are three well-posed research questions which are answered with sufficient analyses, including interpretations of dynamical fields, depiction of wave propagations connecting the tropics to the extratropics, and more elaborated analyses on modal space.

Specific comments:

L130-131: Maybe the authors can further justify their choice of the SST bias center location, amplitude and spatial extent. Perhaps a composite map of the SST bias from those cited studies could be provided, so the readers can see how much the idealized SST bias used in this study resembles that in the coupled models. This may help place the experimental design in better context with existing knowledge.

The SST biases used in this study are not the same as in the coupled climate models in terms of the center location, spatial extent and magnitude, but they resemble those in the coupled models.

The SST biases used in EXP_POS and EXP_IOD have direct correspondences in CMIP5 models. Figure R1 shows ensemble mean CMIP5 bias in climatological SST by month. The SST biases in November and December are similar, which extend almost the whole tropical Indian Ocean along the equator with a maximum magnitude of about 1 K. We use the SST perturbation in EXP_POS to mimic this bias. The SST biases from June to October have a major positive center in western tropical Indian Ocean and several negative centers in Bay of Mengal and around Indonesia. The maximum amplitude is about 1.5 K. This bias is represented by the dipolar SST bias in EXP_IOD.

As for the SST biases used in EXP_NEG and EXP_10N, they are mainly used to study the sensitivity of the response to the sign and meridional location of the SST bias, respectively. But note that the CMIP5 SST has negative bias in the Arabian sea and the Bengal sea from January to May, and the maximum negative amplitude is about -2 K. The SST bias in EXP_10N is somewhat similar to this bias, but with the opposite sign.

To address your concern, we added more references and explanations to the choice of the SST bias. See Lines 149-152 in the revised ms:

"These SST perturbations are similar to, but not the same as, the SST biases in the coupled climate models in terms of the center location, spatial extent and magnitude. The ones used in EXP_POS and EXP_IOD have their counterparts in CMIP5 models (see Fig. 4 in Lyon et al. (2020)). Those in EXP_NEG and EXP_10N are primarily used to study the sensitivity of the response to the sign and meridional location of the SST bias, respectively."

[Figure]

Figure R1. Multimodel mean CMIP5 bias (K) in climatological SST (1979–2005) by month (Fig. 4 in Lyon et al. (2020))

L250-263: Maybe also briefly describe Figs 3c and 3d.

Done. We have added the following sentences to the text in Lines 261-263:

"The northward shift of the SST bias does not have much impact on the spatial structure of the precipitation biases, but reduces the magnitude (Fig. 3c). As for the dipolar SST bias, it causes much stronger precipitation biases in the TIO region (Fig. 3d) than monopolar SST biases (Figs. 3a-3c)."

L275-276: Maybe further elaborate on how they should account for the extratropical precipitation biases in Fig 3.

This is simply due do the upward (downward) motions in the extratropical cyclonic (anticyclonic) circulation biases. For instance, in EXP_POS, one can see a cyclonic circulation bias and an anticyclonic circulation bias over the north Pacific (Fig. 4b in the ms), accompanied by positive and negative precipitation biases respectively (Fig. 3a in the ms).

We have added the following sentences to the text (Lines 281-283):

"For instance, in EXP_POS, one can see a cyclonic circulation bias and an anticyclonic circulation bias over the North Pacific (Fig. 4b), accompanied by positive and negative precipitation biases, respectively (Fig. 3a). "

L277-278: Maybe suggest why the northward shift doesn't seem to have a large impact on the circulation biases.

This is likely associated with the background state. Previous studies have shown that SST changes in tropical ascending regions is more efficient at generating global influences (e.g., Zhou et al 2017). In DJF, the Hadley cell ascending branch is located slightly south of the equator. (This also explains the south-north asymmetry in the precipitation biases in the TIO region in EXP_POS (see Fig. 3a in the ms).) Therefore, when the SST bias shifts northward away from the ascending region, it becomes less efficient at producing atmospheric response.

We added a short discussion in Lines 285-288.

"… albeit with smaller amplitude. Previous studies have shown that SST changes in tropical ascending regions are more efficient at generating global impacts (e.g., Zhou et al. 2017). In DJF, the ascending branch of the Hadley circulation is located slightly south of the equator. Thus, as SST bias moves northward away from the equator, it becomes less efficient at impacting the atmosphere."

L294: Maybe suggest why the magnitudes are smaller i .e., non-linearity of the response to positive and negative SST biases.

We added a comment to this (Lines 312-313).

"…, implying the nonlinearity of the response to positive and negative SST biases (e.g., Lunkeit and Detten 1997)."

L320-321: The wave in EXP_IOD also seems to terminate earlier before reaching North Africa, unlike the other two experiments. Perhaps this is related to the spatial extent of the jet stream wave-trapping?

The wave in EXP_IOD is trapped by the jet stream as indicated in Fig. 7d in the ms, whereas those in the other experiments are not (Figs. 7a-7c in the ms). Therefore, they have very different routes. The earlier termination of the Rossby waves in EXP_IOD is probably due to the zonal inhomogeneity of the jet stream, which is weak to the west coast of North America and is not conducive to Rossby wave propagation.

The following sentence is added to the text (in Lines 342-344):

"The termination of the Rossby waves over America in EXP_IOD is probably due to the zonal inhomogeneity of the jet stream, which is very weak to the west coast of North America and does not support Rossby wave propagation."

L360: To understand this energy budget, the reader needs to refer to equation 16. Maybe consider citing the equation here again to facilitate understanding. Similarly, equation 17 for L371.

Done. See Lines 379, 384, and 401. Note that the equation numbers have changed in the revision.

L364-366: This seems to explain why I and E both decrease together in Figs 9c and 9d for EXP_IOD. Taking a step back, what is the reason of why I and E decrease in EXP_IOD but increase in the other three experiments?

This has been explained in Lines 419-421:

"$\Delta E$ in EXP_IOD is dominated by $k = 1 - 3$ and is mostly attributed to the P term which has relatively large negative values at $k = 1$ and 2 (Fig. 10c). This could be related to the large phase differences (greater than $\pi/2$) between the climatological Walker circulation and the biases caused by dipolar SST bias (see Supplement).

This information has been incorporated in the Conclusions (Lines 494-497):

"They increase in experiments with a monopolar SST bias and decrease in the case with a dipolar SST bias. The increase is mainly due to bias variance, while the decrease is due to a strong negative covariance between the bias in unbalanced circulation and the reference state (characterized by Walker circulation) at zonal wavenumbers 1 and 2."

and in the Abstract (Lines 17-19):

"Both increase in experiments with monopolar SST bias and decrease in that with dipolar SST bias. The increase is mainly due to the bias variance, whereas the decrease is due to a strong negative bias covariance at zonal wavenumbers $k = 1$ and 2."

L367: Similarly, why do I and E decrease in all experiments except EXP_NEG where they increase in Fig 9e?

This is related to different scenarios of the nonlinear wave-zonal mean flow interaction in experiments with positive and negative SST biases. The wave circulation biases caused by SST biases modify the zonal-mean flow, which in turn alters the nonlinear interaction between them. We demonstrate this by comparing EXP_POS and EXP_NEG. Figure 2R shows the changes in the DJF baroclinic energy transfer (BC) and barotropic energy transfer (BT) from zonal-mean flow to wave flow (see Zhao and Liang (2018) for the energy transfer formula) in the extratropics with respect to the reference simulation. One can see that BC dominates the changes in both experiments. It is overall weakened in EXP_POS and strengthened in EXP_NEG. Therefore, the energy (I and E) decreases in EXP_POS and increases in EXP_NEG.

A relevant example can be found in Zhao and Liang (2018) on the interannual variation of the wintertime North Pacific jet stream and storm tracks, where the strong and weak jet streams correspond to two different energy scenarios.

We added the following sentences to the text (in Lines 394-398):

"This implies a different scenario of the nonlinear wave-zonal mean flow interaction in EXP_NEG from the others. The wave circulation biases caused by SST biases can modify the zonal-mean flow, which in turn alters the nonlinear interaction between them. It is found that the biases in EXP_NEG act to strengthen the baroclinic energy transfer from zonal-mean flow to wave flow, leading to wave energy increase (not shown)."

[Figure]

Figure R2. Changes in the DJF baroclinic energy transfer (BC; $m^2$ $s^{-3}$) and barotropic energy transfer (BT; $m^2$ $s^{-3}$) from zonal-mean flow to wave flow in the extratropics with respect to the reference simulation.

L374-376: While the indication from Fig 9 is clear, the readers may also wonder if there is an explanation behind it.

Explanations are provided in Lines 387-389 (and the Supplement):

"The decrease in the unbalanced zonal-mean energy should be attributed to the weakening of the Hadley circulation, whereas the increase in the balanced zonal-mean energy is due to the diabatic heating induced by positive SST bias and the strengthening of the westerlies in midlatitudes and the easterlies in the tropics (see Supplement)."

and in the Conclusions (Lines 489-491):

"These changes primarily arise from the covariance between the circulation bias and the reference state, which results in a weakening of the Hadley circulation in the unbalanced regime and a strengthening of the zonal-mean westerlies (easterlies) in the midlatitudes (tropics) in the balanced regime."

This information has been incorporated in the Abstract (Lines 15-16):

"These changes primarily arise from the strong covariance between the circulation bias and the reference state (i.e., bias covariance)."

L380-382 and L473-475: Again, the readers may wonder if there is an explanation for why EXP_NEG increases V.

The temporal variability response is more difficult to understand than its spatial counterpart. But a comment on the extratropical variance changes can be provided. It is found that the energy and variance in the extratropics show an in-phase response across all experiments (Fig. 9e in the ms), unlike the tropics (Fig. 9d in the ms). This is reminiscent of the definition of mid-latitude storm tracks, which can be quantified by temporal variance V (e.g., Blackmon et al. 1977) or storm energy E (e.g., Zhao and Liang 2019), and they are equivalent. It implies that the variance and energy are closely related in midlatitudes. This is not the case in the tropics, which is a bit confusing.

L479-486: The authors have highlighted the limitations of this study. It would be also useful if the authors could highlight the implications of this study on the interpretation of CMIP5 and CMIP6 model results, based on the results from this intermediate complexity model.

Thanks for your suggestion. We add the following sentence to the text (in Lines 509-510):

"Our paper provides a novel dynamical framework for evaluating simulations of CGCMs, such as CMIP models, that have suffered from severe SST biases over time."

Technical corrections:

L55: tropical-extratropical coupling

Changed to "tropics-extratropics coupling".

L107: Clausius

Corrected.

L135: specifies the longitude and latitude of the center location respectively

Modified.

L282: are yet to be fully understood

Corrected.

Fig 8a: maybe label the horizontal axis.

The abscissa indicates the value range of the Z profile, which is dimensionless, so no labelling is required.

L347: do the authors mean Figs 8c and 8d instead?

Corrected.

L348: do the authors mean Figs 8e and 8f instead?

Corrected.

Fig 9 caption: maybe briefly remind the readers what the symbols used in the horizontal axis stand for, which were described in Section 2.

The caption has been modified. Thanks.

**Thanks again for your careful review!**

Yuan-Bing Zhao and Coauthors

References:

Blackmon, M. L., J. M. Wallace, N.-C. Lau, and S. L. Mullen, 1977: An Observational Study

of the Northern Hemisphere Wintertime Circulation. *J. Atmos. Sci.*, **34**, 1040–1053, https://doi.org/10.1175/1520-0469(1977)034<1040:AOSOTN>2.0.CO;2.

Lunkeit, F., and Y. von Detten, 1997: The Linearity of the Atmospheric Response to North Atlantic Sea Surface Temperature Anomalies. *Journal of Climate*, **10**, 3003–3014, https://doi.org/10.1175/1520-0442(1997)010<3003:TLOTAR>2.0.CO;2.

Lyon, B., 2020: Biases in CMIP5 Sea Surface Temperature and the Annual Cycle of East African Rainfall. *Journal of Climate*, **33**, 8209–8223, https://doi.org/10.1175/JCLI-D-20-0092.1.

Zhao, Y.-B., and X. S. Liang, 2018: On the Inverse Relationship between the Boreal Wintertime Pacific Jet Strength and Storm-Track Intensity. *J. Climate*, **31**, 9545–9564, https://doi.org/10.1175/JCLI-D-18-0043.1.

——, and ——, 2019: Causes and underlying dynamic processes of the mid-winter suppression in the North Pacific storm track. *Sci. China Earth Sci.*, **62**, 872–890, https://doi.org/10.1007/s11430-018-9310-5.

Zhou, C., M. D. Zelinka, and S. A. Klein, 2017: Analyzing the dependence of global cloud feedback on the spatial pattern of sea surface temperature change with a Green's function approach. *J. Adv. Model. Earth Syst.*, **9**, 2174–2189, https://doi.org/10.1002/2017MS001096.

---

## Author Comment (AC4)

Reviewer 2

Dear referee,

Thanks a lot for your thorough and insightful review. The following is a point-to-point reply. The answers are in blue.

The authors investigated the multidecadal atmospheric bias teleconnections caused by the TIO SST bias and their impacts on the simulated atmospheric variability using an intermediate-complexity atmospheric model. The authors found that the atmospheric circulation biases caused by the TIO SST bias have a Gill-Matsuno-type pattern in the tropics and a Rossby wave-train distribution in the extratropics. They also showed that the TIO SST bias could influence interannual variations in the tropical Indo-West Pacific region, Australia, south and northeast Asia, the Pacific-North America region, and Europe.

It is important to understand how the tropical SST bias in the model could generate atmospheric teleconnection biases. I have read the manuscript with much interest. It was especially interesting to see the response of the teleconnections.

The paper is well-written and well-organized. Thus, I suggest the work be accepted subject to some minor revisions.

L230–235: Focusing on the boreal winter is fine. However, if you want to say "Indeed, the circulation bias in the tropics has the same pattern throughout the year, only with varying magnitude, and the extratropical biases are primarily observed in the Northern Hemisphere during boreal winter and in the Southern Hemisphere during boreal summer, albeit with much weaker intensity", please show it.

Sorry for the inaccurate statement. Figures R1 and R2 give the unbalanced and balanced wave circulation biases, respectively. We can see that the wave circulation bias in the tropics (25°S-25°N) follows a similar pattern throughout the year but with varying magnitude. The unbalanced fields are dominated by the Kelvin mode. The balanced fields are featured as a quadrupole structure in the tropics. The extratropical biases are primarily observed in the Northern Hemisphere during boreal winter and in the Southern Hemisphere during boreal summer. In transition seasons (spring and autumn), extratropical biases exist in both Hemispheres with similar magnitude.

We rephrased the sentences in Lines 235-238:

"Indeed, the circulation bias in the tropics (25°S-25°N) has similar pattern throughout the year but with varying magnitude, and the extratropical biases are primarily observed in the Northern Hemisphere during boreal winter and in the Southern Hemisphere during boreal summer. In transition seasons (spring and autumn), extratropical biases exist in both Hemispheres with similar magnitude."

[Figure]

Figure R1. Unbalanced circulation biases at σ = 0.211. Vectors stand for winds (in m s⁻¹) and colors for the geopotential height (in gpm). Black dots mark the SST bias centers.

[Figure]

Figure R2. Balanced circulation biases at σ = 0.211. Vectors stand for winds (in m s⁻¹) and colors for the geopotential height (in gpm). Black dots mark the SST bias center.

The IO has strong seasonality. So, I expected the seasonality to be critical. If you focus only on the boreal wintertime, the authors should add "boreal wintertime" in the title.

Agree.

We changed the title to "Atmospheric bias teleconnections **in boreal winter** associated with systematic SST errors in the tropical Indian Ocean".

The abstract was modified accordingly (Lines 5-6):

"Bias teleconnections **with a focus on boreal wintertime** are researched using …"

L137–145: Please add some references to explain why you set the EXP and EXP_10N as similar to EXP_IOD.

We added an explanation to the choice of the SST bias. See Lines 149-152 in the revised ms:

"These SST perturbations are similar to, but not the same as, the SST biases in the coupled climate models in terms of the center location, spatial extent and magnitude. The ones used in EXP_POS and EXP_IOD have their counterparts in CMIP5 models (see Fig. 4 in Lyon et al. (2020)). Those in EXP_NEG and EXP_10N are primarily used to study the sensitivity of the response to the sign and meridional location of the SST bias, respectively."

Fig. 7: Why not show EXP_10N? I expect that EXP_10N may influence RWS and WAF more than EXP_POS/NEG.

Figure R3 shows the WAF and RWS in each experiment. We can find that the impact of EXP_10N SST bias on the RWS and WAF is not as strong as that of EXP_POS. This is likely associated with the background state. Previous study has shown that SST changes in tropical ascending regions is more efficient at generating global influences (Zhou et al 2017). In DJF, the Hadley cell ascending branch is located slightly south of the equator. (This also explains the south-north asymmetry in the precipitation biases in the TIO region in EXP_POS (see Fig. 3a in the ms).) Therefore, when the SST bias shifts northward away from the ascending region, it becomes less efficient at producing wave sources and therefore the WAF.

Another major difference between EXP_POS and EXP_10N is seen over Asia and north Pacific. In EXP_POS, there are two wave paths in that region. One (the northern path) originates in west and east Asia and propagates northeastward and then eastward, and the other (the southern path) originates in the subtropical North Pacific and propagates northeastward. They merge over the northeast Pacific. In EXP_10N, however, only the north path is visible.

In the revised ms, the EXP_10N has been added to Figure 7. Besides, we added a short discussion in Lines 285-288:

"… albeit with smaller amplitude. Previous studies have shown that SST changes in tropical ascending regions are more efficient at generating global impacts (Zhou et al. 2017). In DJF,

the ascending branch of the Hadley circulation is located slightly south of the equator. Thus, as SST bias moves northward away from the equator, it becomes less efficient at impacting the atmosphere."

[Figure]

Figure R3. Horizontal distribution of the 250 hPa stationary wave activity flux (vectors, $m^2\ s^{-2}$) and Rossby wave source (colors, $10^{-11}\ s^{-2}$) in DJF: (a) EXP_POS, (b) EXP_NEG, (c) EXP_10N, and (d) EXP_IOD. Black contours show the balanced geopotential height biases. The contour interval is 5 gpm. Negative values are indicated with dashed lines and the zero-line is omitted.

And we rephrased the text in Lines 334-344 accordingly:

"The major feature is the wave propagation indicated by WAF over Aisa and the PNA region. In EXP_POS, there are two wave paths. One (the northern path) originates in Asia and spreads northeastward and then eastward; the other (the southern path) originates in the subtropical North Pacific and propagates northeastward. The two wave paths merge over the

northeast Pacific and then propagate eastward across North America and the North Atlantic, and finally terminates over North Africa (Fig. 7a). EXP_NEG has similar wave propagation to EXP_POS, but its northern wave path is very weak (Fig. 7b). In EXP_10N, the northern wave path is similar to that of EXP_POS, but the southern wave path no longer exists (Fig. 7c). The wave-train in EXP_IOD originates in South Asia. It first spreads northeastward and then eastward across the North Pacific (Fig. 7d). The wave route in EXP_IOD is much zonal, which may be due to the wave being trapped by the jet stream (Zhang and Liang, 2022). The termination of the Rossby waves over America in EXP_IOD is probably due to the zonal inhomogeneity of the jet stream, which is very weak to the west coast of North America and does not support Rossby wave propagation."

The authors analyzed the years 1931–2010. As the authors may know, the Indian Ocean has warmed faster than the global average. So I believe that atmospheric bias teleconnections associated with IO SST bias may change. Although the topic is beyond the main scope, it might be better to slightly touch on the problem.

Thank you for pointing out this.

The atmospheric bias teleconnections associated with IO SST bias may have a time evolution with the non-stationary backround SST, but it is beyond the scope of this study. A brief discussion does not help to draw any conclusive results. We plan to study this in a separate paper.

In response to your suggestion, we added the following sentences in Lines 520-522.

"Furthermore, the TIO has warmed faster than any other tropical oceans over the past century (Roxy et al. 2020), and the atmospheric bias teleconnections associated with the TIO SST bias may have a temporal evolution with the non-stationary background SST. While not addressed in this paper, it will be discussed in follow-on studies."

**Thanks again for your careful review!**

Yuan-Bing Zhao and Coauthors

References:

Lyon, B., 2020: Biases in CMIP5 Sea Surface Temperature and the Annual Cycle of East African Rainfall. *Journal of Climate*, **33**, 8209–8223, https://doi.org/10.1175/JCLI-D-20-0092.1.

Roxy, M. K., and Coauthors, 2020: Indian Ocean Warming. *Assessment of Climate Change over the Indian Region: A Report of the Ministry of Earth Sciences (MoES), Government of India*, R. Krishnan, J. Sanjay, C. Gnanaseelan, M. Mujumdar, A. Kulkarni, and S. Chakraborty, Eds., Springer, 191–206.

Zhou, C., M. D. Zelinka, and S. A. Klein, 2017: Analyzing the dependence of global cloud feedback on the spatial pattern of sea surface temperature change with a Green's function approach. *J. Adv. Model. Earth Syst.*, **9**, 2174–2189, https://doi.org/10.1002/2017MS001096.